# CORTICO-CEREBELLAR NETWORKS AS DECOUPLED NEURAL INTERFACES

## ABSTRACT

The brain solves the credit assignment problem remarkably well. For credit to be correctly assigned across multiple cortical areas a given area should, in principle, wait for others to finish their computation. How the brain deals with this locking problem has remained unclear. Deep learning methods suffer from similar locking constraints both on the forward and backward phase. Recently, decoupled neural interfaces (DNI) were introduced as a solution to the forward and backward locking problems. Here we propose that a specialised brain region, the cerebellum, helps the cerebral cortex solve the locking problem closely matching the computations and architecture of DNI. In particular, we propose that classical cerebellar forward and inverse models are equivalent to solving the backward and forward locking problems, respectively. To demonstrate the potential of this framework we focus on modelling a given brain area as a recurrent neural network in which the cerebellum approximates temporal feedback signals as provided by BPTT. We tested the cortico-cerebellar-DNI (CC-DNI) model in a range of sensorimotor and cognitive tasks that have been shown to be cerebellar-dependent. First, we show that the CC-DNI unlocking mechanisms can facilitate learning in a simple target reaching task. Next, by building on the sequential MNIST task we demonstrate that these results generalise to more complex sensorimotor tasks. Our cortico-cerebellar model readily applies to a wider range of modalities, to demonstrate this we tested the model in a cognitive task, caption generation. Models without the cerebellar-DNI component exhibit deficits similar to those observed in cerebellar patients in both motor and cognitive tasks. Moreover, we used CC-DNI to generate a set of specific neuroscience predictions. Finally, we introduce a CC-DNI model with highly sparse connectivity as observed in the cerebellum, which substantially reduces the number of parameters while improving learning through decorrelation. Overall, our work offers a novel perspective on the cerebellum as a brain-wide decoupling machine for efficient credit assignment and opens a new avenue of research between deep learning and neuroscience.

## 1 INTRODUCTION

Efficient credit assignment in the brain is a critical part of learning. However, how the brain solves the credit assignment problem remains a mystery. One of the central issues of credit assignment across multiple stages of processing is the need to wait for previous stages to finish their computation before others can proceed (Rumelhart et al., 1986; Schmidhuber, 1990; Lee et al., 2015; Marblestone et al., 2016; Jaderberg et al., 2017). In deep artificial neural networks these constraints are explicit. During the forward phase a given layer has to wait for all its previous layers to finish before it can proceed, a constraint known as the *forward lock*. Similarly, during the backward phase a given layer has to wait for all the layers above to finish computing its gradients – *backward lock*. Recently, a framework was introduced to decouple artificial neural networks – decoupled neural interfaces (DNI; (Jaderberg et al., 2017))[1], effectively breaking forward and/or backward locks.

Here, we propose that a specialised brain area, the cerebellum, performs a similar role in the brain. In the classical view the cerebellum is key for fine motor control and learning by constructing internal models of behaviour. (Marr, 1969; Albus, 1971; Raymond and Medina, 2018; Wolpert et al., 1998;

---

[1] DNIs are related to earlier work on using network critics to train neural networks (Schmidhuber, 1990).

Miall et al., 1993). More recently, however, the idea that the cerebellum is also involved in cognition has gained significant traction (Schmahmann et al., 2019; Wagner and Luo, 2020; Brissenden and Somers, 2019). An increasing body of behavioural, anatomical and imaging studies points to a role of the cerebellum in cognition in humans and non-human primates (Schmahmann et al., 2019; Brissenden and Somers, 2019; Guell et al., 2015; 2018). Impairments in cerebellar patients occur across a range of tasks including language (Guell et al., 2015), working memory (Deverett et al., 2019), planning (Baker et al., 1996), and others (Fiez et al., 1992). These observations suggest that the cerebellum implements a *universal function* across the brain (Marr, 1969; Albus, 1971; Raymond and Medina, 2018; Diedrichsen et al., 2019). Moreover, experimental studies looking at cortico-cerebellar interactions have demonstrated that cerebellar output is crucial for maintaining neocortical representations in order to drive behaviour (Chabrol et al., 2019; Gao et al., 2018). However, to the best of our knowledge, no theoretical framework has considered what might be the function of such interactions between the cerebellum and cortical areas.

In an attempt to reduce the existing gap between experimental observations and existing computational approaches we introduce DNI as a cortico-cerebellar model – *cortico-cerebellar DNI* (CC-DNI). Consistent with the cerebellar universal role we theorise that the cerebellum serves to break the locks inherent to both feedforward and feedback information processing in the brain, akin to DNI. In particular, we posit that the two classical internal models of the cerebellum, forward and inverse models, are equivalent to DNI-mediated unlocking of feedback (gradients) and feedforward communication, respectively. Following this view the cerebellum not only provides motor or sensory estimates, but also any other modality encoded by a particular brain region. Inspired by neuroscientific studies, we test our model on sensorimotor tasks: (i) a target reaching task (Sanes et al., 1990; Butcher et al., 2017; Nashef et al., 2019) and (ii) a set of more complex temporal tasks based on the MNIST dataset, but also (iii) on a cognitive task – caption generation (Guell et al., 2015). Our results support the cortico-cerebellar DNI models we study and show that they generally speed up learning by unlocking the main network, qualitatively consistent with a wide range of behavioural observations (Guell et al., 2015; Sanes et al., 1990; Butcher et al., 2017; Nashef et al., 2019).

Two defining features of the cerebellum are the large expansion at the granule cell input layer with 50 billion neurons (the most numerous cell in the brain) and the highly sparse connectivity (each granule cell receives $\sim 4$ synapses) (Sanger et al., 2020). These observations have been long suggested to help speed-up learning in the cerebellum through decorrelation (Albus, 1971; Sanger et al., 2020; Cayco-Gajic et al., 2017). Building on these studies we introduce a new DNI model, *sparse CC-DNI*. Consistent with classical cerebellar models (Albus, 1971; Cayco-Gajic et al., 2017) we show that input sparsity can improve learning in the presence of high correlations. We finish with a discussion on the implications and predictions of this new brain-wide model of the cerebellum.

## 2 CEREBELLUM AS A DECOUPLING MACHINE

We first describe DNIs following Jaderberg et al. (2017) and then establish the link to cortico-cerebellar networks. Assume that a feedforward neural network consists of $N$ layers, with the $i$th layer ($1 \leq i \leq N$) performing a "computational step" $f_i$ with parameters $\theta_i$. Given input $x$ at layer 1, the output of the network at its final layer is therefore given by $f_N(f_{N-1}(\ldots f_2(f_1(x))\ldots))$. We use $\mathcal{F}_i^j$ to denote the composition of steps from layer $i$ to layer $j$ (inclusively). Finally, let $h_i$ denote the (hidden) activity at layer $i$, so that $h_i = f_i(h_{i-1})$ with $h_0 = x$.

To illustrate the locking constraints of standard artificial neural networks used in deep learning suppose that a network is in the process of learning via backpropagation, with current input-target pair $(x, y_{\text{targ}})$. To update the layer parameters $\theta_i$ the gradient $\frac{\partial L}{\partial \theta_i}$ is required, where $L = \mathcal{L}(y, y_{\text{targ}})$ is the loss which compares the target value against the model output $y = \mathcal{F}_1^N(x)$ under some loss function $\mathcal{L}$; we then apply gradient descent on the parameters, $\theta_i \leftarrow \theta_i - \alpha \frac{\partial L}{\partial \theta_i}$, with learning rate $\alpha > 0$. Suppose however that the network has only recently received the input $x$ and is currently only at module $i$ of the forward computation. In order to update the corresponding parameters of that layer, $\theta_i$, the layer must first wait for all remaining layers to finish $f_j$ ($j > i$) for the loss to be computed. Only then the various gradients of the loss are backpropagated and $\frac{\partial L}{\partial \theta_i}$ is finally available.

These two characteristics of backpropagation make layer $i$ "backward locked" to $\mathcal{F}_{i+1}^j$, enforcing a strong dependence of the layer's learning to the speed of forward and backward propagation through

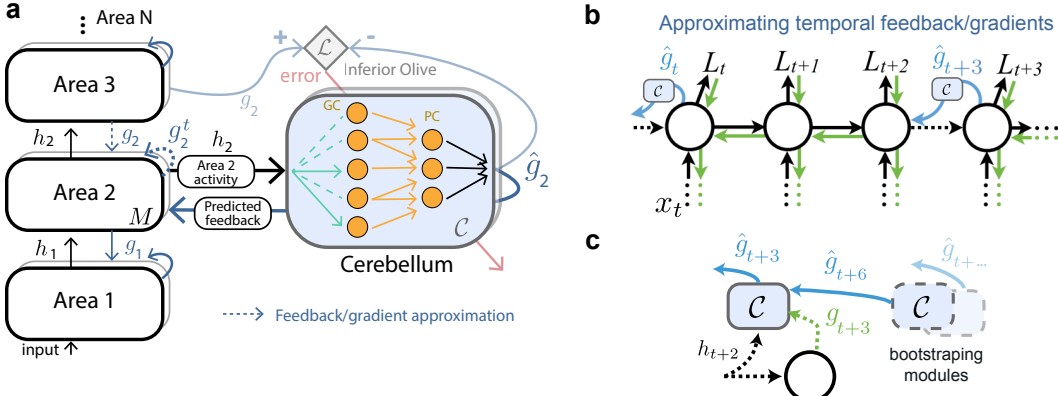

Figure 1: Cortico-cerebellar networks as decoupled neural interfaces (CC-DNI). (**a**) The cerebellum approximates forward (black arrows) or feedback (blue arrows) cortical processing, effectively decoupling the respective brain areas as done by DNI Jaderberg et al. (2017). Here we highlight the case of unlocking feedback processing. The cerebellum attempts to approximate feedback signals ($\hat{g}_2 = \{g_2^t, g_2\}$) which can be purely spatial ($g_2$) or temporal $g_2^t$. This means that area 2 does not need to wait for the exact feedback signals. The cerebellum receives neural activity from area 2, which project (sparsely) through mossy fibers (cyan) onto granule cells (PC; orange) and Purkinje cells (PC; orange). Cerebellar learning is mediated by the inferior olive, which compares estimated feedback $\hat{g}_2$ with real feedback $g_2$, computing $\mathcal{L}_\mathcal{C} = ||g_2 - \hat{g}_2||$ (see main text). Sparse connectivity onto GCs (connections and pruned connections denoted by solid and dashed cyan lines, respectively) represents the sparseDNI model. (**b**) Illustration of the cerebellum approximating temporal feedback signals provided by BPTT. By approximating future feedback (e.g. at $\hat{g}_t$ and $\hat{g}_{t+3}$) the cerebellum reduces the need for strong BPTT. Adapted from Jaderberg et al. (2017). (**c**) To be trained with temporal feedback the cerebellum bootstraps its own teaching signals. This is consistent with the modular structure of the cerebellum, where the each module's predictions are used by the cerebellum itself to speed up cerebellar learning.

the rest of the network. Similarly the the network is forward locked during the forward pass. DNI can be used to unlock both backward and forward locks Jaderberg et al. (2017).

To illustrate the model here we focus on *backward DNI*, which goal is to break the backward lock by feeding the hidden layer $i$ activity $h_i$ to a separate neural network, a backward synthesiser $\mathcal{C}_i^B$, that learns to produce a *synthetic gradient* $\hat{g}_i$ – an estimate of the real gradient expected by layer $i$, $\mathcal{C}_i(h_i) = \hat{g}_i \approx \frac{\partial L}{\partial h_i}$. This synthetic gradient can then be used to update the weights of layer $i$ as soon as $h_i$ is available, following

$$\theta_i \leftarrow \theta_i - \alpha_\mathcal{C} \hat{g}_i \frac{\partial h_i}{\partial \theta_i}; \qquad \hat{g}_i = \mathcal{C}_i(h_i) \tag{1}$$

More specifically, the parameters of $\mathcal{C}_i^B(h_i)$ are learned by comparing the estimated gradient with a target gradient $\bar{g}_i$ so as to minimise $\mathcal{L}_\mathcal{C} = ||\bar{g}_i - \hat{g}_i||$. Ideally we set the target gradient as the true gradient, $\bar{g}_i = \frac{\partial L}{\partial \theta_i}$, as can be implemented without difficulty in the case of feedforward networks. However, if we consider the temporal case (see sections 2.1.1 and 3) where we wish to estimate gradients of future losses many timesteps ahead, $L = \sum_t L_t$, the true backpropagated gradient is computational expensive to obtain. Jaderberg et al. (2017) counter this potential problem by applying a bootstrapping principle in which a DNI module itself is used to guide learning; explicitly, the target gradient at timestep $t$ is $\bar{g}_t = \sum_{\tau > t}^T \frac{\partial L_\tau}{h_t} + \mathcal{C}_T(h_T) \frac{\partial h_T}{\partial h_t}$, where $T$ defines some limited horizon; note the interesting resemblance to the $n$-step return used in reinforcement learning algorithms.

## 2.1 CORTICO-CEREBELLAR DNI MODELS

Building directly on DNI we introduce a model of cortico-cerebellar computation (CC-DNI). These models use a simple feedforward neural network, consistent with the mostly feedforward architecture of the cerebellum (Marr, 1969; Albus, 1971; Raymond and Medina, 2018). The input layer of the

cerebellum module $\mathcal{C}$ models the mossy fiber input onto granule cells (GCs; Fig. 1), which are the most numerous neurons in the brain ($> 50$ billion in humans (Herculano-Houzel, 2009)). Consistent with this cerebellar dimensionality expansion, in our models we use $M \gg N$, where $M$ is the number of GCs and $N$ the number of neurons in the main (cortical) network (Fig. 1). In particular, we use ratios $\frac{M}{N} \sim 4$, consistent with experimental observations (Herculano-Houzel, 2009). This is different to DNI, in which the synthesizer uses a single hidden layer with the same number of units as LSTM (for comparison in performance see ref to Fig. S2). In addition, the hidden layers and the output of $\mathcal{C}$ approximate the role of GCs and Purkinje cells, and the cerebellar output nuclei, respectively.

Cerebellar granule cells receive sparse input connections ($K$) with only around 3-7 synapses per GC (Herculano-Houzel, 2009; Eccles et al., 2013). These architectural constraints have led to sparse encoding and decorrelation theories of cerebellar function (Albus, 1971; Cayco-Gajic et al., 2017; Schweighofer et al., 2001; Broomhead and Lowe, 1988; Billings et al., 2014; Litwin-Kumar et al., 2017; Sanger et al., 2019). Inspired on these features, we introduce a new model – *sparse CC-DNI* (sCC-DNI), for which we set a small number of incoming input connections $K = 4$ (Fig. 1).

To measure decorrelation achieved by sCC-DNI we use the Pearson correlation and a population correlation metric $r_{\text{pop}}$ that has been shown to better capture cerebellar effects (Cayco-Gajic et al., 2017) $r_{\text{pop}} = \frac{Z}{Z-1} \left( \frac{\max\{\sqrt{\lambda_i}\}_i}{\sum_i \sqrt{\lambda_i}} - \frac{1}{Z} \right)$ where $Z$ are the number of neurons being considered and $\lambda_i$ are the eigenvalues of the covariance matrix of the neuronal activity (e.g. $h_M$). To manipulate the hidden correlations of the model, we adjust the variance of its input and recurrent weights by scaling each by a factor $b$ with $b \neq 1$ (see SM).

There is a strong similarity between CC-DNI models and the flow of information in standard internal models of the cortico-cerebellar networks (Fig. 1, Table S1). Below we draw a parallel between classical cerebellar internal models and our CC-DNI models. For simplicity, below and in our results we focus on the link between forward internal models and backward DNI (but see S1 for our interpretation of inverse internal model as forward DNI).

### 2.1.1 Forward cerebellar models as backward DNI

We propose that classical forward cerebellar models are equivalent to backward DNI. In the forward model of sensorimotor control, the cerebellum receives an efferent copy of the motor command from the motor cortex (Miall et al., 1993; Ito, 1970), and sensory feedback from motor centres. With these two inputs the forward model learns to predict the sensory consequences of motor commands. We argue that a similar predictive model can be applied to predict cerebral activity in brain regions such as the prefrontal cortex and the temporo-parietal cortex which are involved in planning of cognitive behaviour and decision making (Schmahmann et al., 2019; Wagner and Luo, 2020; Brissenden and Somers, 2019; Ito, 2008) (Fig. 1a). Similarly to the forward model backward-DNI also receives an efferent copy, which is represented by the hidden activity from a given layer (or brain area) $h_i$. From this input the synthesiser learns to predict the gradient with respect to the activity of layer $h_{i+1}$ (Fig. 1a,c). In addition, we suggest that the cost function is computed by the *inferior olive* (e.g. $\mathcal{L} = ||g_M - \hat{g}_M||$; Fig. 1a), which mediates learning in the cerebellum via climbing fibres, consistent with existing cerebellar theoretical frameworks (Marr, 1969; Albus, 1971; Raymond and Medina, 2018). Finally, the prediction $\hat{g}_M$ is sent back to the brain.

Such backward DNIs are of particular importance when using recurrent neural networks (RNNs), which learn to estimate future gradients from timestep $t + 1$ given current state at time $t$ (Jaderberg et al., 2017) (Fig. 1b; see SM for more details). Our tasks use RNNs with weak (i.e. with low truncations) backpropagation through time (BPTT) as it allow us to more clearly demonstrate the potential of the CC-DNI models. As is the case in DNIs the cerebellum relies on a bootstrapped cost function (Fig. 1c). We interpret such bootstrapping as being provided by the current or other cerebellar modules, consistent with the highly modular cerebellar architecture (Apps et al., 2018). Here we use LSTMs (Hochreiter and Schmidhuber, 1997) as a model of cortical networks, which can be potentially linked to cortical microcircuit (Costa et al., 2017).

### 2.1.2 Encoding gradients in the brain

Recent developments have introduced biologically plausible solutions to how the brain encodes gradients (Lillicrap and Santoro, 2019; Guerguiev et al., 2017; Sacramento et al., 2018; Richards

and Lillicrap, 2019; Payeur et al., 2020; Ahmad et al., 2020). Regarding spatial backpropagation of gradients (i.e. across layers), Sacramento et al. (2018) demonstrated that biological networks do not need to explicitly send gradient information, but rather that this can be reconstructed locally in dendritic microcircuits. On the other hand Bellec et al. (2019) showed that temporal gradients as used in BPTT can be approximated by eligibility traces that transmit information forward in time. Both of these solutions can be incorporated into our framework, in which CC-DNI would predict feedback activity originating from upstream brain areas (as in Sacramento et al. (2018)) and/or eligibility traces in RNNs (see S2 for a schematic; Bellec et al. (2019)). However, this is outside of the scope of the current paper and we leave this to future work (see more detailed discussion in SM).

## 2.2 COMPARISON WITH EXISTING CEREBELLAR COMPUTATIONAL MODELS

Over the past decades the regular circuitry of the cerebellum has driven theorists to model the cerebellar cortex in order to understand its function (Kaiser et al., 2018). The Marr-Albus models are particular popular models which view the local cerebellar circuitry as multi-layer perceptrons (Albus, 1971). Another modelling framework that has been used is based on Kalman filters, which are also used to model the local circuitry to generate sensorimotor predictions (Porrill et al., 2013). These models have enabled significant theoretical and experimental advances. However, they do not consider potential interactions with the wider brain, in particular the neocortex, which are known to exist and appear to be critical for cerebellar function (Gao et al., 2018). Moreover, these models have so far only captured relatively simplistic (sensorimotor) tasks (e.g. eye-blink conditioning). In contrast, our framework provides a solution to both of these issues, while being consistent with existing local models of the cerebellum.

## 3 RESULTS

In this paper we focus our experiments on backward-CC-DNI models (cerebellar forward model; Fig. 1a,b) which we test on a range of task domains that have been shown to be cerebellar-dependent. Below we provide a summary of these different tasks and the respective results (see details in SM).

## 3.1 SIMPLE SENSORIMOTOR TASK: TARGET REACHING

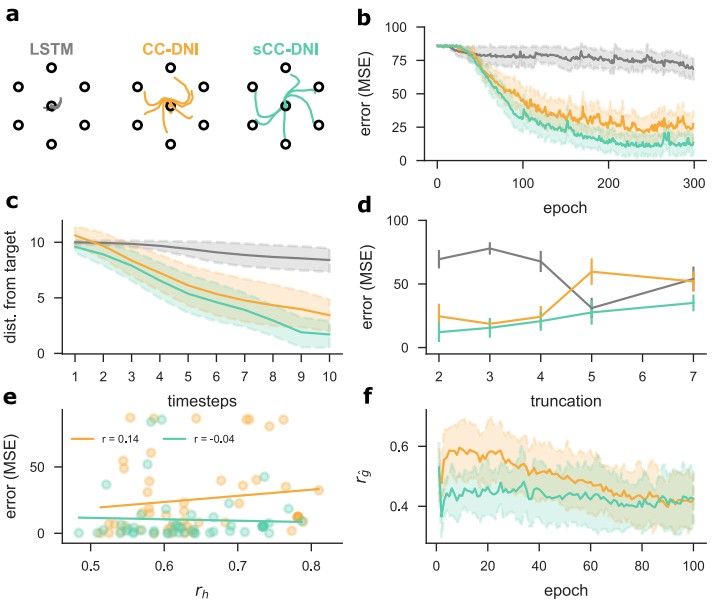

Figure 2: Target reaching task. (**a**) Trajectory produced by LSTM (grey), CC-DNI (orange) and sCC-DNI (cyan) models towards 7 targets when using BPTT $T = 2$. (**b**) Models MSE. (**c**) Distance between model output and the targets. (**d**) MSE for multiple truncations $T$ over the last 5 epochs. (**e**) MSE and correlation of hidden activity ($r_h$) at epoch 1. (**f**) Correlation of CC-DNI estimated gradients ($r_{\hat{g}}$) over the first 100 epochs of learning.

Inspired by classical sensorimotor studies in the cerebellum we first test a simple target reaching task (Sanes et al., 1990; Butcher et al., 2017; Nashef et al., 2019), in which given an input at time $t_1$ the network needs to reach a target (Fig. 2a and S8; see SM for more details). In this task error signals are only available at the end of task, which must be associated with the initial input. In line

with cerebellar experiments (Sanes et al., 1990; Butcher et al., 2017; Nashef et al., 2019), only the CC-DNI models learn to reach the target with good accuracy, while LSTM still learns but much more slowly (Fig. 2a,b). Interestingly, sparse CC-DNI not only learns more quickly than CC-DNI (Fig. 2b,d, and S3), but also reaches the target more quickly (i.e. in fewer time steps) than the two other models (Fig. 2c). Consistent with cerebellar theories (Sanger et al., 2020; Cayco-Gajic et al., 2017; Billings et al., 2014) sCC-DNI helps when high correlations are present in the main network activity $h$ (Fig. 2e), due to decorrelation of its estimated gradients $\hat{g}$ (Figs. 2f; see S6 for pop. corr.). CC-DNI properties are particularly important when learning with low BPTT truncations (Fig. 2d), predicting that the cerebellum is particularly important for more temporally challenging tasks and consistent with DNI theory (Jaderberg et al., 2017).

## 3.2 ADVANCED SENSORIMOTOR TASKS

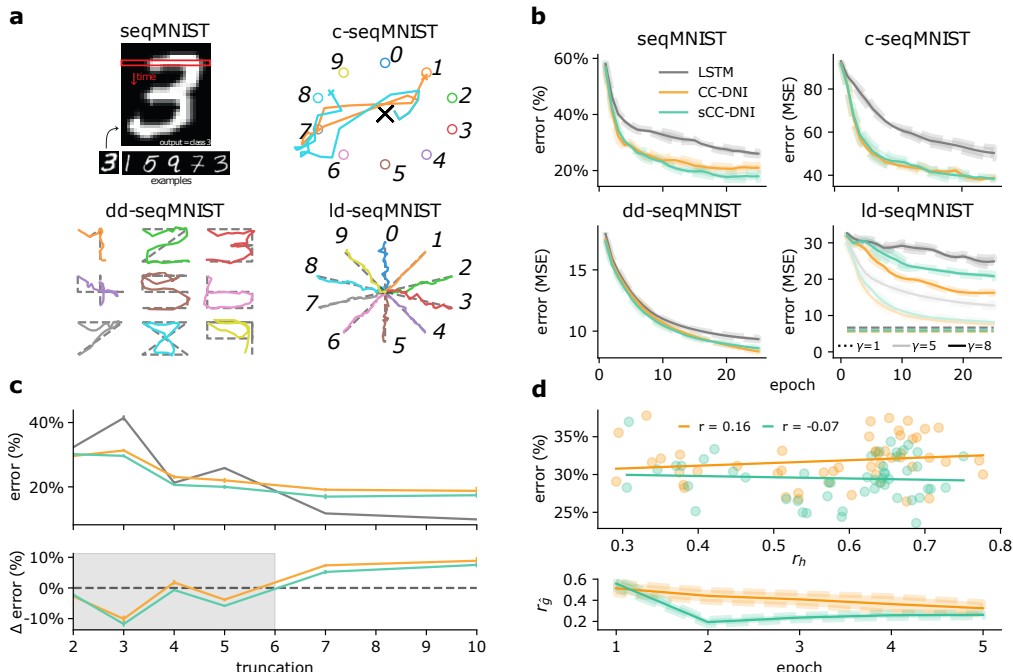

Figure 3: Sequential MNIST tasks. (**a**) Illustration of the four variants of sequential MNIST (seqMNIST). Top left: Standard classification sequential MNIST (row-by-row). Top right: given seqMNIST input the model is asked to reach a target coordinates (c-seqMNIST); Bottom left: given seqMNIST input the model is asked to draw the digit following a template (dd-seqMNIST); Bottom right: as in dd-seqMNIST, but the model is asked to draw a straight line (ld-seqMNIST). Target drawings in dotted grey and model output coloured by digit. (**b**) Validation learning curves for the four tasks with gradient truncation $T = 5$. For ld-seqMNIST, different degrees of loss sparsity $\gamma$ are shown (see main text). (**c**) Model performance across $T$ for seqMNIST (top) and respective differences between DNI and LSTM (bottom). (**d**) Model performance after 5 epochs as a function of the Pearson correlation $r_h$ of the hidden activity at epoch 1 in the main network (dots; top). Correlation of synthetic gradients $r_{\hat{g}}$ produced by CC-DNIs (bottom).

Next, to test whether the target reaching results generalise to more realistic tasks we explored a range of more advanced sensorimotor tasks building on the classical MNIST dataset. We first test the standard sequential MNIST (seqMNIST), in which the model gradually receives an MNIST image row by row before classifying the image as a number $0 - 9$ (Fig. 3a) (Le et al., 2015). As in the target reaching task, the loss function is only defined at the end of the sequence, making this a challenging temporal task.

With truncated BPTT with modest truncation sizes $T \leq 6$, CC-DNI models learn more quickly and achieve better overall performance compared to a standard LSTM (Fig. 3b,c), in line with the previous task. Interestingly, for this task we observe that the cerebellar architecture (i.e. high $\frac{M}{N}$ ratio, and sparse input connections $K$) to be the best solution (Figs. S4, S5). As in the target reaching task,

sparse CC-DNI helps in the case of high correlations in the main network activity $h$, likely achieved by the decorrelation effect sCC-DNI has on the estimated gradients $\hat{g}$ (Figs. 3d, S7).

To see how well the above results generalise to a regression task in which the model has to reach a particular point (as in the target reaching task), we develop a coordinate seqMNIST (c-seqMNIST) task. As before, the model receives the image sequentially, but must now output one out of ten different equally spaced 2D coordinates (Fig. 3a). Note that this is a harder task as models have to learn to transform a complex input pattern towards a specific 2D point (Fig. 3b). Both DNI models outperformed the LSTM model. There is however notably less difference between sCC-DNI and CC-DNI between compared to the standard version of seqMNIST (see also the following tasks), which may suggests that the cerebellum is better suited for pattern separation tasks, in line with existing cerebellar pattern separation theories (Albus, 1971; Sanger et al., 2020; Cayco-Gajic et al., 2017; Litwin-Kumar et al., 2017).

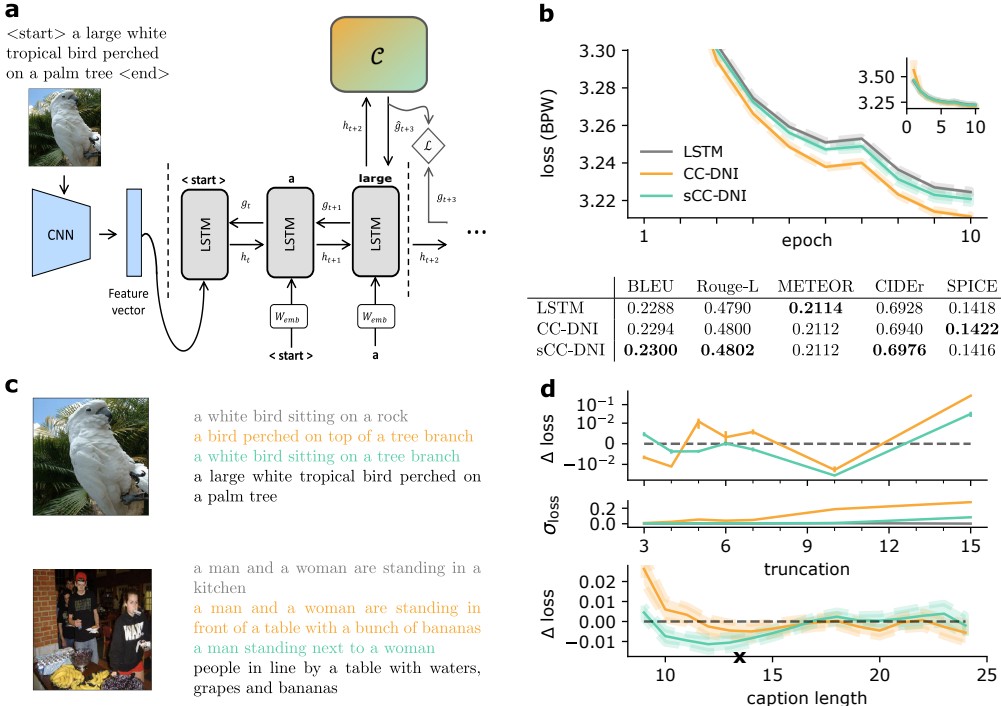

Figure 4: Caption generation task. (**a**) A graphical description of the model. The image is first processed by a pretrained convolutional neural network (CNN). The resulting feature vector is then provided to a connected but distinct RNN which is trained to predict the next word given the previous words of a provided "gold standard" caption to the image. The cerebellum module $\mathcal{C}$ is only applied to the RNN. (**b**) Top: Learning curves in bits per word (BPW) on validation set for gradient truncation $T = 4$. Full learning curve shown as inset. Bottom: standard language metrics (BLEU, Rouge-L, METEOR, CIDEr, SPICE). (**c**) Two example images from the validation set with corresponding model captions and gold standard captions (black). (**d**) Top: Change in loss with respect to LSTM as a function of $T$. Middle: Standard deviation $\sigma_{\text{loss}}$ over the average loss across *all* epochs as a function of $T$. Bottom: Change in loss (i.e. CC-DNIs - LSTM) as a function of caption length, where the cross designates the mean (gold standard) caption length.

Next, we test an explicit drawing task in which we trained the models to draw either a template digit (dd-seqMNIST) or a straight line (ld-seqMNIST) given an image of a digit (Fig. 3a). For these tasks, the loss function is computed at every point in time as $\mathcal{L} = (y_i - \hat{y}_i)^2$, where $y_i$ and $\hat{y}_i$ denote the desired and predicted coordinate at timestep $i$, respectively. Due to the loss being computed at every timestep in both tasks, CC-DNI models are not as important and show a minimal improvement in performance (Figs. 3b, S4). To demonstrate that the temporal resolution of the loss is important for CC-DNI, we varied the loss sparsity, giving the model a loss every $\gamma$ timesteps for ld-seqMNIST. This setup resembles sensorimotor feedback which is typically periodic rather than continuous (Sanes et al., 1990; Synofzik et al., 2008). As expected, sparser losses (higher $\gamma$) improve learning (Fig. 3b).

### 3.3 COGNITIVE TASK: CAPTION GENERATION

Our framework does not only apply to sensorimotor tasks, but should generalise to virtually any task within the grasp of deep learning systems, or indeed the brain. To demonstrate this and inspired by cognitive tasks in which cerebellar patients have shown deficits (Gebhart et al., 2002) we test our models in a caption generation task. In this task the network needs to generate a textual description for a given image. All models have two components: a pretrained convolutional neural network (CNN) to extract a lower dimensional representation of the image, and an LSTM on top to generate text (Fig. 4a). For simplicity and to draw comparison with previous tasks the DNI models contain one synthesiser at the RNN-level, but more could be added to help learning of the CNN.

We use a standard dataset (ILSVRC-2012-CLS (Russakovsky et al., 2015)) and the networks are trained to maximise the likelihood of each target word given an image (SM for more details). We find that CC-DNI models exhibit faster learning (Fig. 4b) for truncation sizes $T \leq 10$ (Fig. 4d) and better generalisation[2] (Fig. S13). All models produce reasonable captions for images unseen during training, but CC-DNI models tend to produce captions more semantically accurate than LSTMs (Figs. 4c, S12), consistent with cerebellar deficits (Guell et al., 2015). In addition, we observe that sCC-DNI are more robust to initial conditions (Fig. 4d), consistent with the decorrelation property. Moreover, consistent with bridging of gradient truncations provided by DNIs, we observe that CC-DNI helps learning in the presence of long captions, whereas sCC-DNI helps mostly for the most common caption lengths (Fig. 4d), suggesting that sCC-DNI better captures the data distribution.

### 3.4 MODEL PREDICTIONS

Our model makes numerous predictions for experimental neuroscience, some of which can be compared with existing observations. First, we tested how important are the exact cerebellar expansion $M/N \sim 5$ and sparsity parameters ($K = 4$) observed experimentally. We find that the parameters found experimentally provide a good combination to facilitate learning in the pattern recognition sequential MNIST task (Fig. 5a). This region of good hyperparameters is much wider in regression tasks such as the coordinate sMNIST (Fig. 5b) or other variants (Fig. S4). This predicts that the cerebellum has evolved to specifically facilitate pattern recognition tasks.

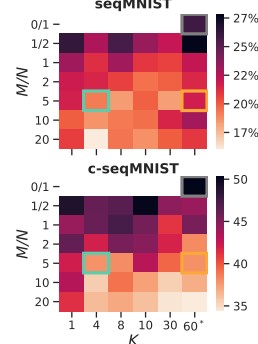

Figure 5: Cerebellar sparsity ($K$) and expansion ($M/N$) parameters in seqMNIST (top) and c-seqMNIST (bottom). Grey, orange and cyan boxes denote LSTM, CC-DNI, sCC-DNI models respectively.

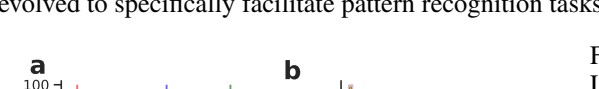
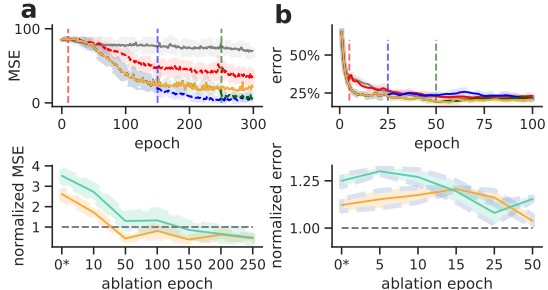

Figure 6: CC-DNI ablation results. (**a**) (top) Learning curve for the target reaching task with multiple total cerebellar ablations at specific epochs (vertical line). (bottom) Summary of average error normalized to the control model, for both CC-DNI (orange) and sCC-DNI (cyan). * denotes LSTM case. (**b**) Same as (**a**) but for seqMNIST.

A common experimental paradigm is to do cerebellar ablation experiments. We used our model to generate specific predictions in terms of behavioural outcome with total cerebellar ablations at different points during learning. We find that ablations generally impair learning (Fig. 6a-b), but that this impairment reduces over learning and once the main network no longer needs the cerebellar estimations its *presence* can impair learning (Fig. 6a-b).

Next, we used our model to make predictions in terms of expected correlations between the neocortex and the cerebellum. Because of the complex temporal gradients in the RNN case here we focus on a

---

[2]See Czarnecki et al. (2017) for discussion on the regularisation effect of synthetic gradients.

simple feedforward network trained on MNIST. In addition, we use this example to demonstrate that when approximating activity directly instead of gradients, as we propose to happen in biology, the cerebellum output converges to some non-zero value (Fig. 7, top). We find that correlations between the main network and the cerebellum increase over learning for neurons with final high correlations whereas correlations decrease for neurons with initial high correlations (Fig. 7, bottom), consistent with recent experimental observations (Wagner et al., 2019). Moreover, we also observe that the variance of total correlations decreases over learning (not shown).

## 4  CONCLUSIONS AND DISCUSSION

We introduced a deep learning model of cortico-cerebellar function, in which the cerebellum acts as a decoupling machine. We demonstrate that this idea is directly related to classical cerebellar models (Marr, 1969; Albus, 1971), but with the key prediction that the cerebellum decouples activity propagation across the brain, speeding up credit assignment across multiple brain areas. Our model is also a step towards solving the temporal credit assignment problem in the brain, as the CC-DNI models reduce the need for strong BPTT (i.e. without trun-

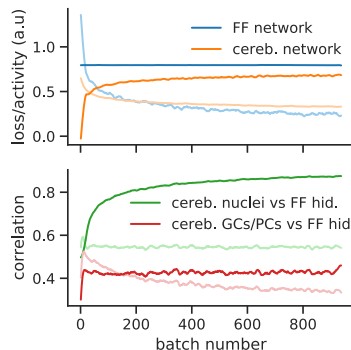

Figure 7: Correlation and activity analysis of feedforward CC-DNI. (top) Training loss (transparent) and mean (absolute) activity (opaque). (bottom) Evolution of the most correlated pairs (256) at beginning (transparent) and end (opaque) of learning.

cations). Temporal credit assignment is directly related to the ability of networks to maintain input information (Bellec et al., 2019). Therefore, our results on truncated BPTT suggest that the cerebellum becomes increasingly important as task difficulty increases due to the inability of cortical networks to maintain a information for longer than hundreds of milliseconds (Deverett et al., 2019).

Our results are largely consistent with observed cognitive deficits in cerebellar patients, such as in language (Guell et al., 2015), which cannot be attributed directly to deficits in motor control (Stoodley and Schmahmann, 2009). Because of the explicit use of labels (or desired targets) here we have relied on supervised learning settings[3], but the same framework can be easily applied in reinforcement and other unsupervised learning settings (Jaderberg et al., 2017), which do not require an explicit teacher. Indeed the cerebellar model we propose here is of particular relevance for reinforcement learning due to the prevalence of sparse and delayed rewards, consistent with recent observations of reward-related signals in the cerebellum (Sendhilnathan et al., 2020).

Our work makes numerous predictions and opens the exciting possibility of comparing CC-DNI models with recent studies across a range of tasks (King et al., 2019; Ashida et al., 2019). We performed three separate experiments to highlight predictions made by the model: (i) we used ablations to show how behavioural performance is shaped by cerebellar predictions, (ii) a study on cerebellar expansion and sparsity that suggests that the cerebellum may have evolved to facilitate learning in specific tasks and (iii) study how correlations between main network and the cerebellum develop over learning (see Section 5). Moreover, the model also predicts which connections should project to GCs (source brain area activity) and the inferior olive (target brain area activity). In addition, as demonstrated by Czarnecki et al. (2017) the CC-DNI model also predicts that without a cerebellum the neocortical representations should become less distributed across different brain areas. Furthermore, as shown by Jaderberg et al. (2017) these models enable a network to be trained in a fully decoupled fashion, which means that it can update asynchronously. Given that the brain is asynchronous this may be a fundamental benefit of having a cerebellar system.

Finally, our framework can also inspire new deep learning DNI systems. For example, a key feature of cerebellar architectures are the many functionally separate modules (Apps et al. (2018); Cerminara and Apps (2011); Suvrathan et al. (2016); Fig. 1c) to speed-up learning, generalising the ideas used by Jaderberg et al. (2017).

---

[3]Note, however, that learning of language models (as in caption generation) is often done in an unsupervised (self-supervised) setting.

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

## A  EXPERIMENTAL DETAILS

In each of our tasks we use a long-short term memory network (LSTM; (Hochreiter and Schmidhuber, 1997)) as the main "cortical" network (Costa et al., 2017), and a simple feedforward network of one (sensorimotor tasks) or two hidden layers (caption generation) as the synthesiser "cerebellar" network. As in (Jaderberg et al., 2017) the cerebellar network first predicts gradients for both the memory cell and output state of the LSTM (i.e. for each neuron in the main network the cerebellar network receives two inputs), so that the 'cerebellar' input and output size is two times the number of LSTM units, and this synthetic gradient $\hat{g}$ is then scaled by a factor of 0.1 before being used by the main model for stability purposes. The final 'readout' of the model is a (trained) linear sum of the LSTM output states. All networks are optimised using ADAM (Kingma and Ba, 2014) (see learning rates below).

In each experiment all initial LSTM parameters are drawn from the uniform distribution $\mathcal{U}(\frac{1}{\sqrt{n_{\text{LSTM}}}}, \frac{1}{\sqrt{n_{\text{LSTM}}}})$, where $n_{\text{LSTM}}$ is the number of LSTM units. Other than the final layer, as in (Jaderberg et al., 2017), the feedforward weights of the cerebellar network are initialised according to $\mathcal{U}(-b_k, b_k)$ where $b_k$ denotes the "kaiming bound" as computed in (He et al., 2015) (slope $a = \sqrt{5}$), and the biases are draw from $\mathcal{U}(\frac{1}{\sqrt{n_{\text{inp}}}}, \frac{1}{\sqrt{n_{\text{inp}}}})$, where $n_{\text{inp}}$ denotes the input size of the layer. As in (Jaderberg et al., 2017), the last layer (both weights and bias) of the cerebellar network is zero-initialised, so that the produced synthetic gradients at the start are zero.

A range of correlations (here quantified as Pearson or population correlation coefficient) for the LSTM activity $h$ can be achieved naturally through different random initialisations (accessed through different random seeds), but can be also controlled by the scaling of initial parameters of the LSTM with a bias factor $b$, where $b > 1$ and $b < 1$ induces greater and smaller variability in parameter values respectively. The results demonstrating the effect of correlation (Figs 2f,g, 3d) are obtained across 10 random seeds, for multiple bias factor values $b \in [0.01, 0.1, 0.5, 1, 2]$.

During learning, backpropagation through time (BPTT) takes place strictly within distinct truncations, and computed gradients are not propagated between them: *truncated* BPTT. As soon as the forward computation of a given truncation is completed, assuming a corresponding (synthetic) loss gradient is available, the model parameters are updated; that is, at a resolution of the preset truncation size, there is *online learning*. We split the original input into truncations as follows. Given an input sequence of $N$ timesteps $x_1, x_2, \ldots, x_N$ and a truncation size $T$, we divide the sequence into $T$ sized truncations with any remainder going to the last truncation. In other words, the sequence is now made up truncations of $(x_1, \ldots, x_T), (x_{T+1}, \ldots, x_{2T}), \ldots, (x_{(m-1)T+1}, \ldots, x_{mT}), (x_{N-r}, \ldots, x_N)$, where $N = mT + r$ for positive integers $m, r$ with $0 \leq r < T$. Note that, along with the value $T$, how well the sequence is divided into truncations (i.e. values $m, r$) is an important factor for learning.

Unless stated otherwise, reported error/loss values are on a held-out validation set at the end of learning. All experiments were conducted using PyTorch.

### A.1  TARGET REACHING TASK

In the target reaching task, an LSTM network receives a discrete input cue which signals the network to move to a particular target location in 2D space. In the results presented in the main text (Fig. 2) we set 6 distinct non-zero input-target pairs $\{(x_i, y_i)\}_{i=1}^{6}$, where each input $x_i$ is a (one dimensional) integer $\in \{\pm 1, \pm 2, \pm 3\}$, and the targets $\{y_i\}_{i=1}^{6}$ lie equidistantly on a circle centred on the origin with radius 10. Once an input cue is received at timestep $t_0$, the model receives no new information (i.e. all future input is zero) and has 10 timesteps to move to the corresponding target location. The model is trained to minimise the mean squared error (MSE) between its final output and the cue-based target. In addition, if no input cue is provided ($x = 0$), the network is trained to not move at all ($y = 0$). Note therefore that only the final model output 'matters' in the sense that it defines model performance.

The cortical network has one hidden layer of 10 LSTM units (i.e. 20 when including the memory cells); unless stated otherwise, the cerebellar network contains one hidden layer

of 80 hidden neurons. The initial learning rate is set to 0.001. Each epoch comprises one batch of 50 randomised examples. Model performance is averaged over 10 random seeds (with error bars), where each seed determines the initial weights of the network. Truncation values $T \in [2, 3, 4, 5, 7]$ are considered.

Furthermore, to see how the above conditions generalise we consider variants of the task with 1. different numbers of cue/target pairs (see Fig. S8a and b) and 2. higher dimensional input (see Fig. S8c and d). More specifically, we consider two cases where the model receives two-dimensional (Fig. S8c) and 28-dimensional (Fig. S8d) input. For the former the model architecture remains the same except now the model receives at time $t_0$ the (two-dimensional) target values scaled by a factor 0.1 as input. For the latter we have an LSTM with 30 hidden units and the input $x$ is now a binary vector of size 28. In order to increase the task difficulty for these variants of higher dimensional input, we add to the input Gaussian noise with variance 0.1 at each timestep.

### A.2 SEQUENTIAL MNIST TASKS

For each seqMNIST based task the model receives the same temporal MNIST input, and the tasks are only differentiated by the model output. Given a $28 \times 28$ MNIST image, at timestep $i$ the model receives the pixels from row $i$ of the image, so that there are 28 total timesteps and an input size of 28. Only the final model output matters for the seqMNIST and c-seqMNIST tasks - that is, the loss is solely defined at the end of the sequence - whereas the model output is assessed at every timestep for dd-seqMNIST and ld-seqMNIST.

In each case we have one hidden layer of 30 LSTM units in the main model and one hidden layer of 300 (unless stated otherwise) hidden units in the feedforward cerebellar network. Data was presented in batches of 50 with an initial learning rate of 0.0001.

Training and validation data was assigned a $4 : 1$ split, containing 48000 and 12000 distinct image/number pairs respectively. The truncation values $T \in [2, 3, 4, 5, 7, 10]$ are considered. Model performance averaged (with error bars) over 3 random seeds for weight initialisation.

#### A.2.1 SEQMNIST

This is the standard form of seqMNIST, where at the end of the presentation of the image the model must classify in the image as a number between 0 and 9. The output of the model is a vector probabilities of size 10 (one entry for each number), and the model was trained to maximise the likelihood of the correct number.

#### A.2.2 C-SEQMNIST

In this variant each number 0-9 MNIST image is allocated an $xy$ position on the edge of a circle centred at 0, with the position uniquely determined by the digit represented by the image (Fig. 3a). With the model output then a vector of size 2, the training loss is defined at the end by the mean squared error (MSE) between the final output of the model and the target coordinate; since the radius of the circle is length 10 (arbitrary unit), a static model, where the model output remains at 0, would have a constant MSE of 100.

#### A.2.3 DD-SEQMNIST

Like c-seqMNIST, in this variant the model outputs 2D coordinates, but now the model must learn to predict an entire sequence of coordinates $\{\hat{y}_i\}_{i=1}^{28}$. The target sequence $\{y_i\}_{i=1}^{28}$ can be considered a drawing, and in this case resembles the shape of the number itself (Fig. 3c; number 0 not shown). The model is then trained at each timestep to minimise the MSE between the model output at that time and corresponding target point, so that the loss at timestep $t$ defined as $\text{MSE}(y_t, \hat{y}_t)$.

For each number, the corresponding target drawing lies in $[0, 1]^2$, with the gap between each successive point roughly the same. To prevent the model being too harshly judged at timestep 1, all drawings begin in the top left corner $(0, 1)$ (apart from the drawing of 1 which begins slightly beneath/to the right). MSE scores are reported as 100 times their raw values to ease comparison with c-SEQMNIST/dd-SEQMNIST.

#### A.2.4 LD-SEQMNIST

This variant can be thought of as a mixture between c-seqMNIST and dd-seqMNIST. Like dd-seqMNIST the model is constantly assessed and must produce a desired shape, but now

the desired set of points form an equally spaced line where the start point is the origin and the end point is determined by the number-to-coordinate mapping of c-seqMNIST. As with dd-seqMNIST, the loss is defined at each timestep by the difference between the model output and appropriate point on the line.

### A.3 CAPTION GENERATION

The architecture for the caption generation task consists of a pretrained CNN coupled with an RNN (LSTM in this case). The synthesiser (cerebellar network) only communicates to the LSTM. The LSTM network has one layer of 256 LSTM units and the cerebellar network has two hidden layers of 1024 neurons.

The dynamics from image to caption is as follows. As part of image preprocessing and data augmentation, a given image is randomly cropped to size $224 \times 224$, flipped horizontally with even chance, and appropriately normalised to be given to a pretrained resnet model (He et al., 2016). A feature vector $X$ of size 256 is thus obtained and is passed to the LSTM at timestep 0. The LSTM is subsequently presented the "gold standard" caption $\{w_i\}_{i=1}^n$ one word per timestep, each time learning to predict the next word; i.e., at timestep $t$ the model learns $P(w_t|X, \{w_i\}_{i=1}^{t-1})$. The network simultaneously learns a word embedding so that each word $w_i$ is first transformed to a feature vector of size 256 before being served as input. With a preset vocabulary of 9956 distinct words, the final output of the model ($P(w_i)$) is a probability vector of size 9956.

We find that though DNI could help without any explicit method for regularisation (not shown), all models are prone to overfitting. For this reason, we apply dropout (during training) on the input to the LSTM, where a given input element is set to zero with $p = 0.5$ probability.

Once training is complete the models can generate their own unique captions to previously unseen images (Figs. 4, S12). Given an image at timestep 0,, the model applies a 'greedy search' where the output of the model at timestep $i$ is the word with the highest probability, and the same word is then provided as input to the model at timestep $i + 1$. In this way the model eventually outputs an entire sequence of words which forms a caption. In the (highly) rare case where the model generates a sequence of $> 20$ words, we consider only the first 20 words as its caption.

The coco training set ILSVRC-2012-CLS (Russakovsky et al., 2015) holds 414113 total image-caption pairs with 82783 unique images while the held-out validation set holds 202654 with 40504 unique images; note that each image therefore has $\sim 5$ distinct gold standard captions. Training takes place in batches of 100 image/caption pairs, with learning rate of 0.001. Model performance averaged (with error bars) over 5 random seeds for weight initialisation.

In order to judge the models beyond their learning curves in BPW, we quantify their ability to generate captions using a variety of language modelling metrics popular in the realm of language evaluation (image captioning, machine translation, etc). In particular, we compare model-generated captions against the gold standard captions using the metrics BLEU, Rouge-L, METEOR, CIDEr, SPICE (Papineni et al., 2002; Lin, 2004; Denkowski and Lavie, 2014; Vedantam et al., 2015; Anderson et al., 2016). We use BLEU_4 (i.e. consider only $n$-grams for $n \leq 4$) as the default BLEU score (see Fig 4), though we also look at BLEU_1, BLEU_2, BLEU_3 (Fig S15.

## B FORWARD DNI: MNIST

We performed a simple implementation of forward DNI - where now the cerebellar network is trained to predict the activity (not gradients) of the main network - on the MNIST database. With no temporal dimension, the main network is a feedforward neural network which, given a $28 \times 28$ MNIST image, will tune its output (a vector of size 10, where each value corresponds to a possible digit) so as to maximise the probability of the associated number (or equally minimise the cross-entropy loss - Fig. 7, top).

In our case the feedforward network has two hidden layers, each of size 256; after each hidden layer a batch normalisation is applied as well as the non-linear ReLU function. We apply a feedforward cerebellar network with one hidden layer of size 256 which receives

the original MNIST input $x$ and is trained to predict the activity of the second hidden layer of the main network $h_2$; the synthesiser loss used to train the synthesiser parameters then becomes $\mathcal{L}_\mathcal{C} := ||h_2 - \mathcal{C}(x)||$. The cerebellar module can then break the "forward lock" by allowing the main model to bypass its forward computations up until its second layer and instead use $\mathcal{C}(x)$ to make its prediction.

As before, each network is trained using gradient descent with ADAM. Correlations reported are computed using the Pearson correlation. For ease of display reported losses, activity and correlations in Fig. 7 are smoothed over batches using a Savitzky-Golay filter (window size 19, polynomial order 3). Since the output weights of the cerebellar network can be positive or negative, we do not distinguish between positive or anti correlation (i.e. show the absolute correlation value) between the hidden activity of the cerebellar network and its respective target hidden activity of the feedforward network.

## C  Comparison with other models of the cerebellum

In section 2 of the main text we argue that there is a strong similarity between the flow of information in DNI and classical internal models of the cerebellum. In Table S1 we compare explicitly the input and output components of both types of models.

|  | Forward Model | Backward DNI | Inverse Model | Forward DNI |
|---|---|---|---|---|
| Controller | Neocortex | Main model | Cerebellum | Synthesiser |
| Input | $\mathbf{h}_m$ | $\mathbf{h}_m$ | $\mathbf{h}_s, \mathbf{d}_s$ | $\mathbf{h}_s$ |
| Output | $f_{\text{unk}}(\mathbf{h}_m)$ | $\mathcal{C}^B(\mathbf{h}_m) = \frac{\partial \hat{L}}{\partial \mathbf{h}_m}$ | $f_{\text{unk}}(\mathbf{h}_s, \mathbf{d}_s)$ | $\mathcal{C}^F(\mathbf{h}_s)$ |
| Ouput destination | Neocortex | Main model | Controlled object | Main model |

Table S1: Internal models versus DNI. The properties of the forward model of the cerebellum can be set against those of backward DNI (blue); similarly, the properties of the inverse model of the cerebellum can be set against those of forward DNI (red). Notation is largely consistent with section 2 of the main text: $\mathbf{h}_m, \mathbf{h}_s$ denotes the hidden activity of a motor area and sensory area, respectively; $\mathcal{C}^B, \mathcal{C}^F$ denotes the computation of backward DNI and forward DNI, respectively; $L$ denotes the loss function. In addition, the inverse model of the cerebellum traditionally also has access to a *desired* state $\mathbf{d}_s$ (in particular, one can consider this a special case of "context" provided to the synthesiser; cf (Jaderberg et al., 2017)). There are no explicit equations for the computational processes of the forward and inverse models, and both are thus represented by the unknown function $f_{\text{unk}}$.

### C.0.1  Inverse and forward DNI cerebellar models

The inverse model of the cerebellum in motor control computes the motor predictions given a target (or desired) response (Ito, 1970). Feedback errors are derived by comparing a motor command from the motor cortex with the predicted motor command in the cerebellum. We propose that the inverse model is equivalent to forward DNI, as it aims to predict the incoming activity at a given layer (e.g. motor command, $h_M$, or associative input, $h_A$) given the input activity observed at a lower layer (e.g. sensory target response, $h_S$; Fig. 1). Consistent with the modular anatomy of the cerebellum (Apps et al., 2018; Cerminara and Apps, 2011; Ruigrok, 2011) multiple CC-DNI (backward and/or forward) modules can co-exist estimating different pathways components.

## D  Recurrent DNI

In section 2 of the main text we formally describe the dynamics of DNI in a feedforward setting. Here we show how these dynamics easily generalise to the time dependent, recurrent setting. We now take the main model as a recurrent neural network (RNN), where the (recurrent) hidden layer is again connected to a distinct, feedforward synthesiser. By unrolling the network over several timesteps we can keep much of the notation as introduced in section 2, but now $h_i$ denotes the hidden (cortical) activity at *timestep* $i$ (as opposed to layer $i$) with a unique recurrent weight $\theta$ ($= \theta_i \forall i$); furthermore, the loss $L$ is now a sum of individual losses over the timesteps: $L = \sum_i L_i$.

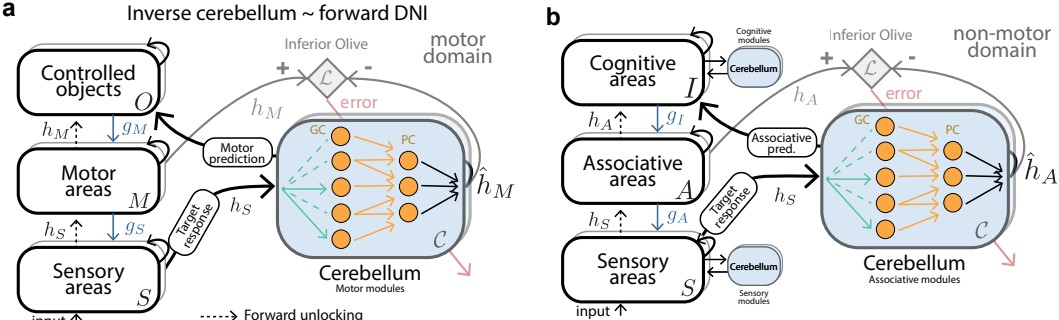

Figure S1: Equivalent to Fig. 1 for how cerebellar inverse models map onto forward DNI (forward-CC-DNI). (**a**) In the inverse model the cerebellum generates motor predictions ($\hat{h}_M$) given desired target responses ($h_S$). This allows the brain to break forward locks (dashed black arrows), for example to go directly from sensory areas to higher areas. The inferior olive plays a similar role to (a,b) but here it allows the cerebellum to learn estimations of forward activities (e.g. $\mathcal{L} = ||h_S - \hat{h}_M||$). (**b**) Example of forward CC-DNI model in a non-motor domain. The cerebellum computes an associative prediction $h_A$ given a sensory input $h_S$. Multiple backward and forward DNI-cerebellum modules can co-exist working in parallel. $h$ and $g$ variables represent forward and gradient information as in DNI (see main text).

DNI is strictly applied to RNNs which learn via truncated BPTT, where BPTT takes place in equally-spaced truncations of time so that any hidden activity will only be attributed to future losses within the truncation window $T$ (i.e. $\frac{\partial L_{i+t}}{\partial h_i} = 0 \ \forall t > T$), effectively ignoring

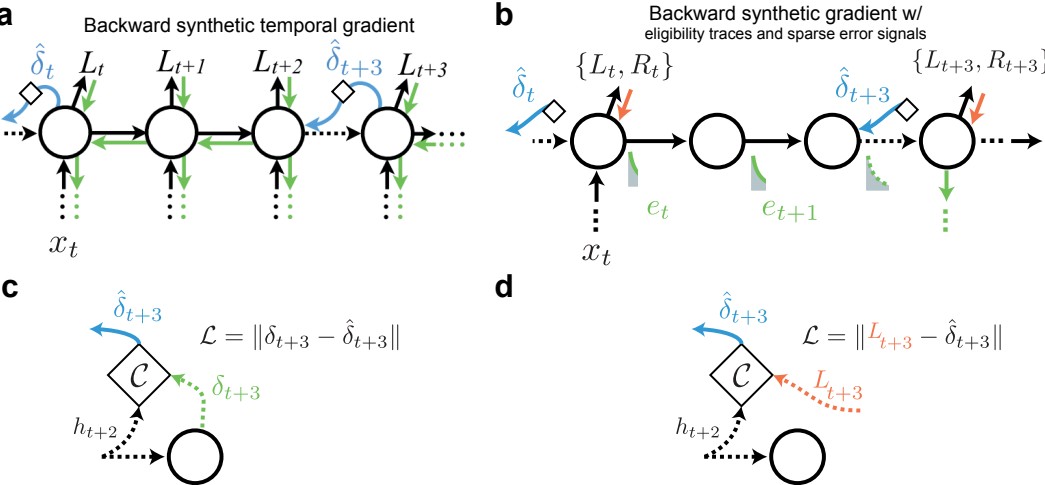

Figure S2: Synthetic gradients in RNNs with eligibility traces. (**a**) An example of backpropagation through time with truncation every T=3 timesteps (dashed black arrows). A DNI can be used to bridge the gap between truncations allowing gradients to be sent further back in time ($\hat{\delta}_t$ and $\hat{\delta}_{t+3}$). Adapted from Jaderberg et al. (2017). (**b**) Schematic of a biologically plausible RNN with eligibility traces that encode information needed to compute the gradients at a later point in time (green traces; cf. (Bellec et al., 2019)). The cerebellar-DNI model $\mathcal{C}$ can be used to estimate gradients $\hat{\delta}$ derived from general losses $L$ or reward signals $R$ to update synaptic weights at a given point in time. Note that teaching signals/rewards are often sparse. (**c**) Schematic of information used by synthesiser to learn in a BPTT setting ($\delta_{t+3}$ represents the future gradients). (**d**) Schematic of information used by synthesiser to learn to predict future losses $\delta$ from the current activity $h$. For biological simplicity we assume that the synthesiser has access to the learning signal directly, but in practice its likely to be a function of $L$.

future losses and potentially disregarding important temporal correlations. The key idea behind recurrent DNI is to bridge these truncations using synthetic gradients (Fig. S2a). For simplicity assume that timestep $i$ is the start of a given truncation. By enabling DNI, a (backward) synthesiser $\mathcal{C}$ is used to provide an estimated gradient $\hat{\delta}_{i+T}$ for all future losses, *outside of the truncation*, with respect to the last hidden activity of the truncation $h_{i+T}$; i.e. $\hat{\delta}_{i+T} \approx \delta_{i+T} = \sum_{j=i+T+1}^{\infty} \frac{\partial L_j}{\partial h_{i+T}}$. The weight update is then analogous to the feedforward case in equation 1

$$\theta \leftarrow \theta - \alpha \Big[ \sum_{j=i}^{i+T} \frac{\partial L_j}{\partial \theta} + \hat{\delta}_{i+T} \frac{\partial h_{i+T}}{\partial \theta} \Big]; \quad \hat{\delta}_{i+T} = \mathcal{C}(h_{i+T}) \tag{2}$$

It is not difficult to see that if the synthesiser perfectly predicts the future gradient, $\hat{\delta}_{i+T} = \delta_{i+T}$, then the weight will be perfectly updated according to the gradient of all future losses, $\theta \leftarrow \theta - \alpha \sum_{j=i}^{\infty} \frac{\partial L_i}{\partial \theta}$. Allowing the synthesiser access to this gold standard, $\delta_{i+T}$, however, requires waiting an arbitrary amount of time and is therefore not practical for training. Instead, $\hat{\delta}_{i+T}$ is trained to replicate an itself approximate, bootstrapped version of the future gradient, $\bar{\delta}_{i+T}$, where $\bar{\delta}_{i+T} = \sum_{j=i+T+1}^{i+2T} \frac{\partial L_j}{\partial h_{i+T}} + \hat{\delta}_{i+2T} \frac{\partial h_{i+2T}}{\partial h_{i+T}}$. Note that $\bar{\delta}_{i+T}$ is available in one truncation's timespan.

In the brain an approximation of BPTT can be implemented forward in time using a mixture of eligibility traces and sparse learning signals (e.g. reward) ((Bellec et al., 2019); Fig. S2b). The dynamics of this more "biologically plausible" learning regime are subtle and are not described here; we simply mention that the cerebellum might be the source of such (future) learning signals and predict that many of the results observed with BPTT will hold for the forward propagation of gradients.

## E  INPUT EXPANSION AND SPARSE CONNECTIVITY IN CEREBELLUM-DNI

We also study the effects of input expansion and sparse connectivity from the main model to the synthesiser in our (s)CC-DNI model. Input expansion and sparse connectivity are two defining features of the cerebellum. Often human intelligence is linked to its expanded neocortex. However, the cerebellar/neocortical cell ratio in humans 4/1 and the cerebellum has significantly expanded throughout evolution (Barton and Venditti (2014); Herculano-Houzel (2010)). Here we aim to study the effect of this input expansion into the cerebellum by systematically varying the number of hidden units in the synthesiser relative to the number of units in the main network. We show this for both sensorimotor task: target reaching task (Fig. S3) and seq-MNIST tasks (Fig. S4 and S5) . In addition, it is known that one granule cell receives input from four mossy fibres on average. Mossy fibres form the main inputs to the cerebellum. The largest source of mossy fibres is the neocortex which projects inputs to the cerebellum via the pons. To quantify the effect of the sparse input connectivity we quantify next to the Pearson correlation also the population correlation coefficient for the target reaching task (Fig. S6) and the standard seq-MNIST task (Fig. S7).

## F  TARGET REACHING SUPPLEMENTARY DATA

Here we show that the target reaching task results extrapolate to variants of the task with varied input-output combinations (Fig. S8). Furthermore, in line with Fig. 2b we include the learning curves across different truncation values for the three models (Fig. S9).

## G  SEQMNIST SUPPLEMENTARY DATA

In line with Fig.3c, we include the learning curves across different truncation values $T$ for the standard seqMNIST task as well as its variants (Figs. S10, S11).

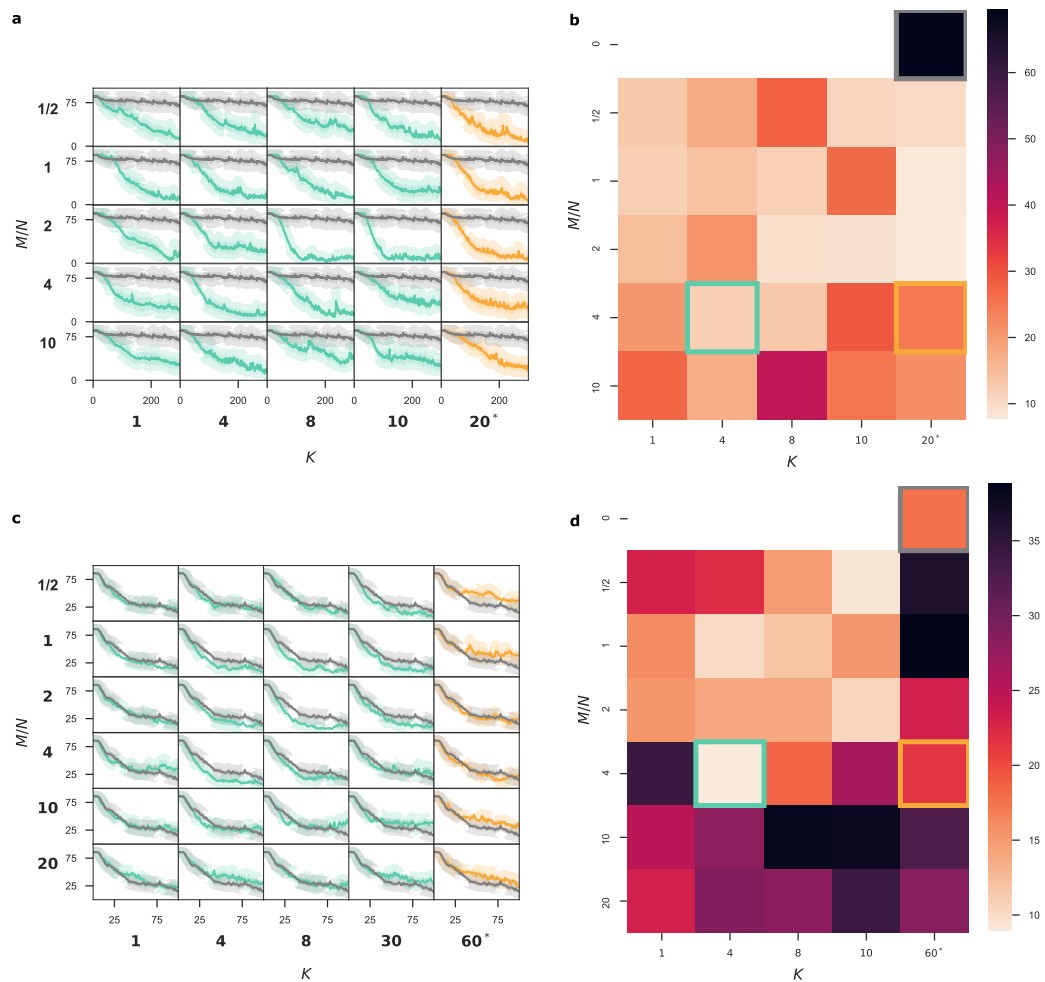

Figure S3: (**a**) Learning curves for target reaching as in Fig. 2 over different divergence ratios $M/N$, where $M$ is the number of hidden 'granular' units in cerebellar model, $N$ the number of input units, and number of non-zero input connections $K$. LSTM performance (grey) shown along with fully (orange) and sparsely (cyan) connected CC-DNI models. Note that $N$ is fixed here as $2 \times 10 = 20$ (10 LSTM units with gradients calculated for both cell and output states), hence * denotes full connectivity. The smaller x-axis in each subplot represents the epoch number and the smaller y-axis represents mean squared error. (**b**) Average performance over last five epochs (295-300) against divergence ratio $M/N$ and input connection sparsity $K$ for target reaching as in Fig. 2. (**c**) Same as (**a**) but for complex input (input dimension =28) and seven 2-D target coordinates (30 LSTM units with gradients calculated for both cell and output states). (**d**) Same as (**b**) for target reaching with complex input (input dimension = 28) and seven 2-D target coordinates over last 5 epochs (95-100).

## H   IMAGE CAPTIONING SUPPLEMENTARY DATA

In line with Fig. 4, we include more examples of image-caption pairs (Fig. S12) as well as the learning curves and metric scores across different truncation values (Figs. S14, S15). Moreover, we demonstrate that DNI models for this task often generalise to unseen data better than regular LSTM models (Fig. S13).

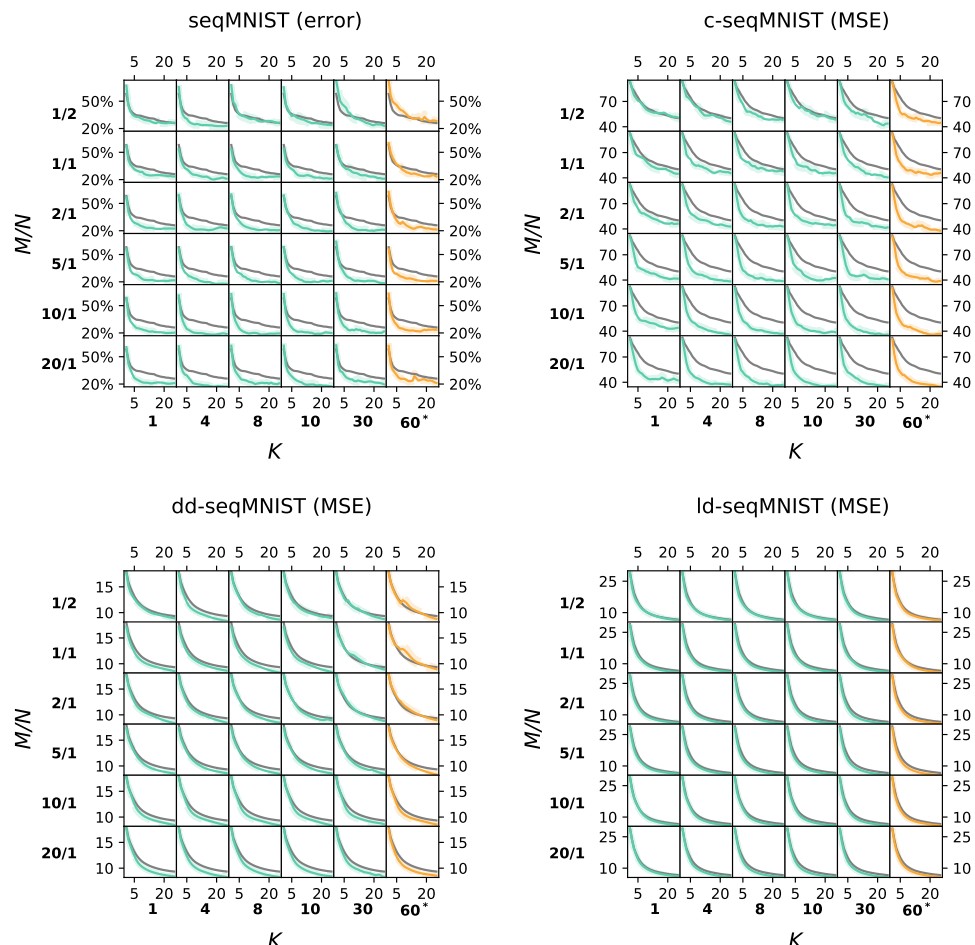

Figure S4: Learning curves for all seqMNIST based tasks over different cerebellum divergence ratios $M/N$, where $M$ is the number of hidden 'granular' units in cerebellar model, $N$ the number of input units, and number of non-zero input connections $K$. LSTM performance (grey) shown as a reference along with fully (orange) and sparsely (cyan) connected cc-DNI models. Note that $N$ is fixed here as $2 \times 30 = 60$ (30 LSTM units with gradients calculated for both cell and output states), hence * denotes full connectivity. The smaller x-axis in each subplot represents the epoch number and the y-axis represents performance over validation data (error for seqMNIST, MSE for the others).

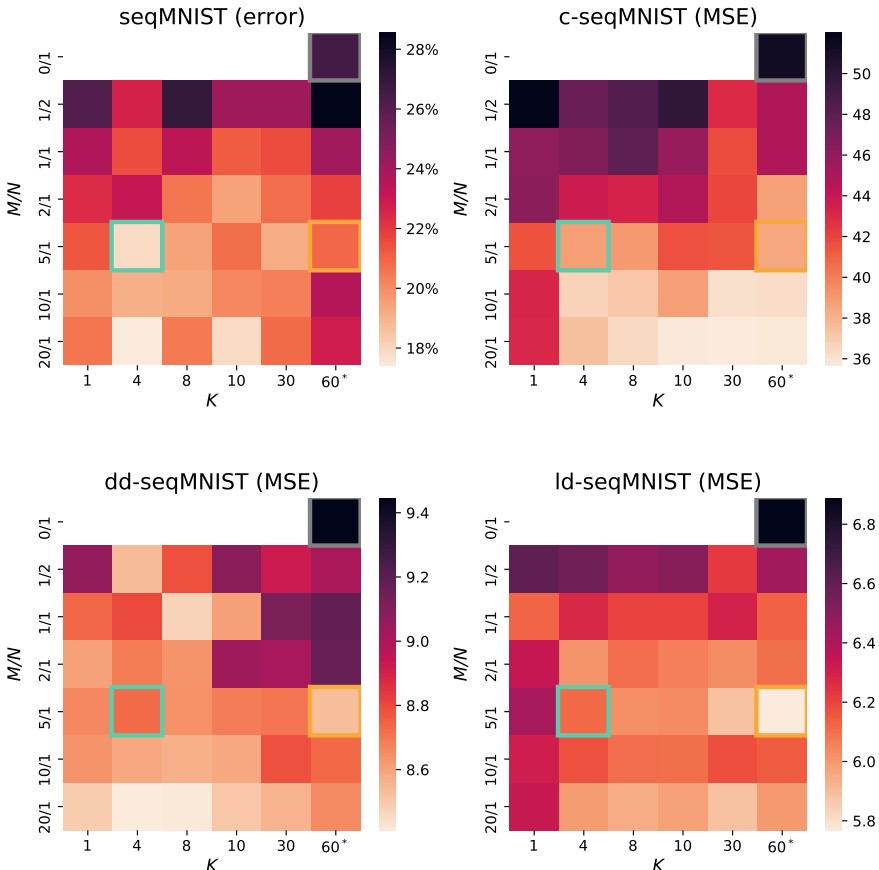

Figure S5: Related to Fig. S4. Average performance over last five epochs (21-25) against divergence ratio $M/N$ and input connection sparsity $K$ for each seqMNIST based task. * denotes full connectivity. The divergence ratio and connectivity of the default LSTM, CC-DNI and sCC-DNI models used for the seqMNIST tasks (see Fig. 3) are illustrated by the grey, orange and cyan squares respectively.

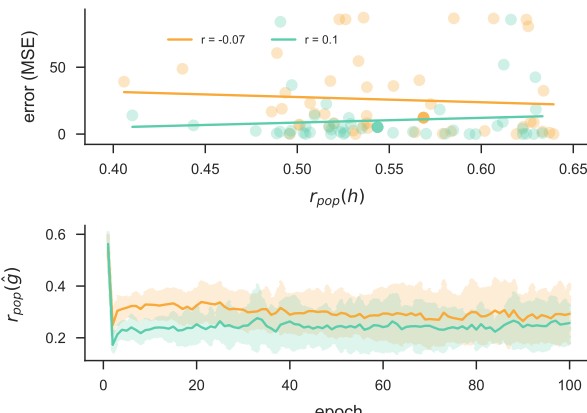

Figure S6: Equivalent to Fig. 2f for population correlation $r_{\mathrm{pop}}$ (see equation **??**). (top) Effect of correlation of hidden (cortical) activity $h$ on performance for cc-DNI (orange) and sCC-DNI (cyan) on the target reaching task, where population correlation and performance are recorded during the first and last (300) epoch respectively. (bottom) Evolution of the population correlation of the synthetic gradient $\hat{g}$ over the first hundred epochs.

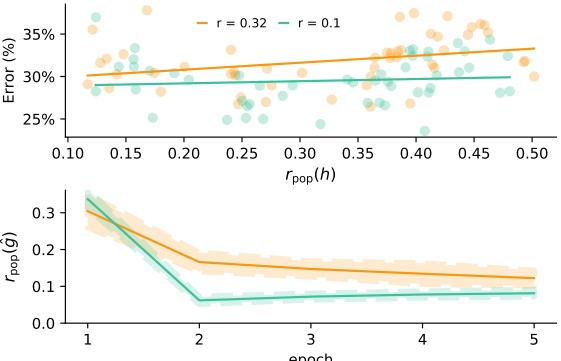

Figure S7: Equivalent to Fig. 3d for population correlation $r_{\mathrm{pop}}$ (see equation **??**). (top) Effect of correlation of hidden (cortical) activity $h$ on performance for cc-DNI (orange) and sCC-DNI (cyan) on the (standard) seqMNIST task, where population correlation and performance are recorded during the first and fifth epoch respectively. (bottom) Evolution of the population correlation of the synthetic gradient $\hat{g}$ over the first five epochs.

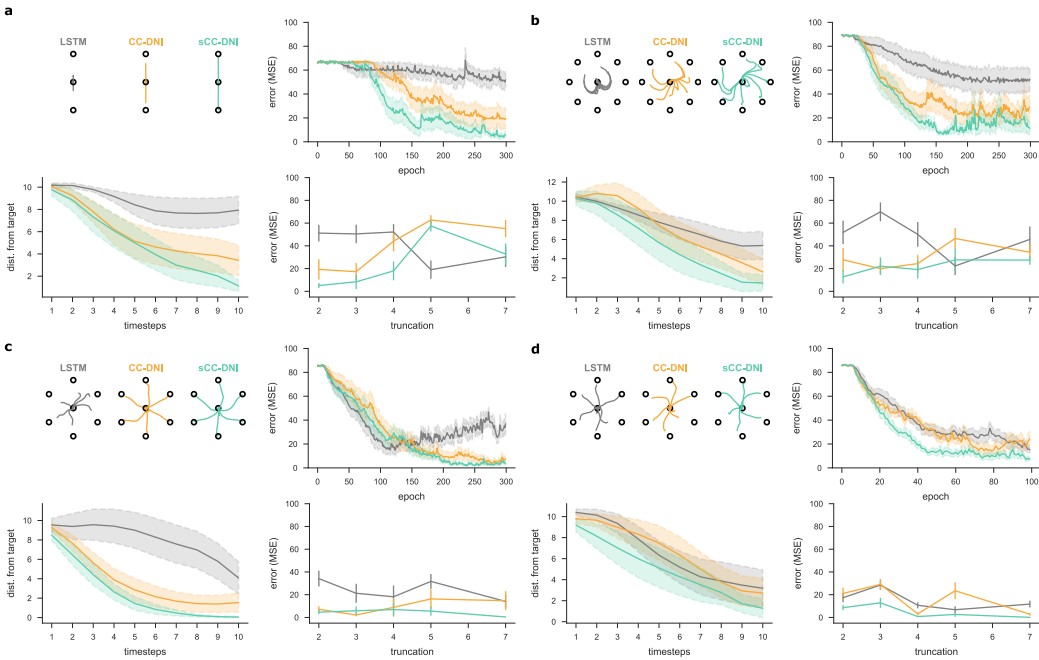

Figure S8: (**a, b**) Target reaching task as in Fig. 2 with varying number of targets. (**a**) Given a one-dimensional input the network has to learn to drive its output towards one of the 3 targets over 10 timesteps. For this variant divergence ratio for CC-DNI is $M/N = 4$ and $K = 4$ for sCC-DNI. (**b**) Given a one-dimensional input the network has to learn to drive its output towards one of the 9 targets over 10 timesteps. For this variant divergence ratio for CC-DNI is $M/N = 2$ and $K = 4$ for sCC-DNI. (**c, d**) Two variants on the target reaching task with varying input dimension, two-dimensional (i.e. $X = \{x_1, x_2\}$) and and 28-dimensional (Fig. S8d) (i.e. $X = \{x_1, ..., x_28\}$) input respectively. (**c**) Given a two-dimensional input the network has to learn to drive its output towards one of the 7 targets over 10 timesteps. For this variant divergence ratio for CC-DNI is $M/N = 4$ and $K = 4$ for sCC-DNI. (**d**) Given a 28-dimensional input the network has to learn to drive its output towards one of the 7 targets over 10 timesteps. For this variant divergence ratio for CC-DNI is $M/N = 4$ and $K = 4$ for sCC-DNI.

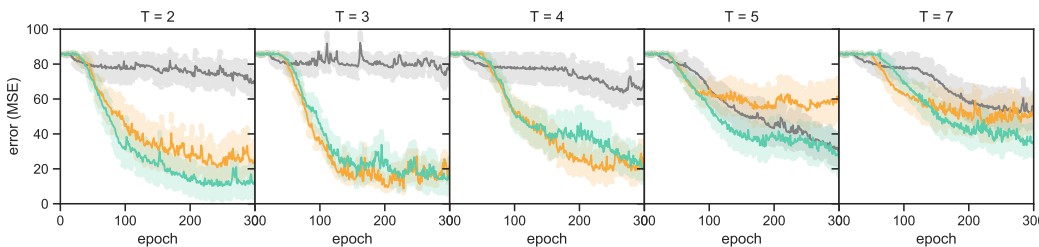

Figure S9: Learning curves for the target reaching task as in Fig. 2b across different truncation values $T$ shown for the three models LSTM (grey), CC-DNI (orange) and sCC-DNI (cyan). Parameters are the same as in Fig. 2b with divergence ratio $M/N = 4$ and sparse input connectivity $K = 4$ for sCC-DNI.

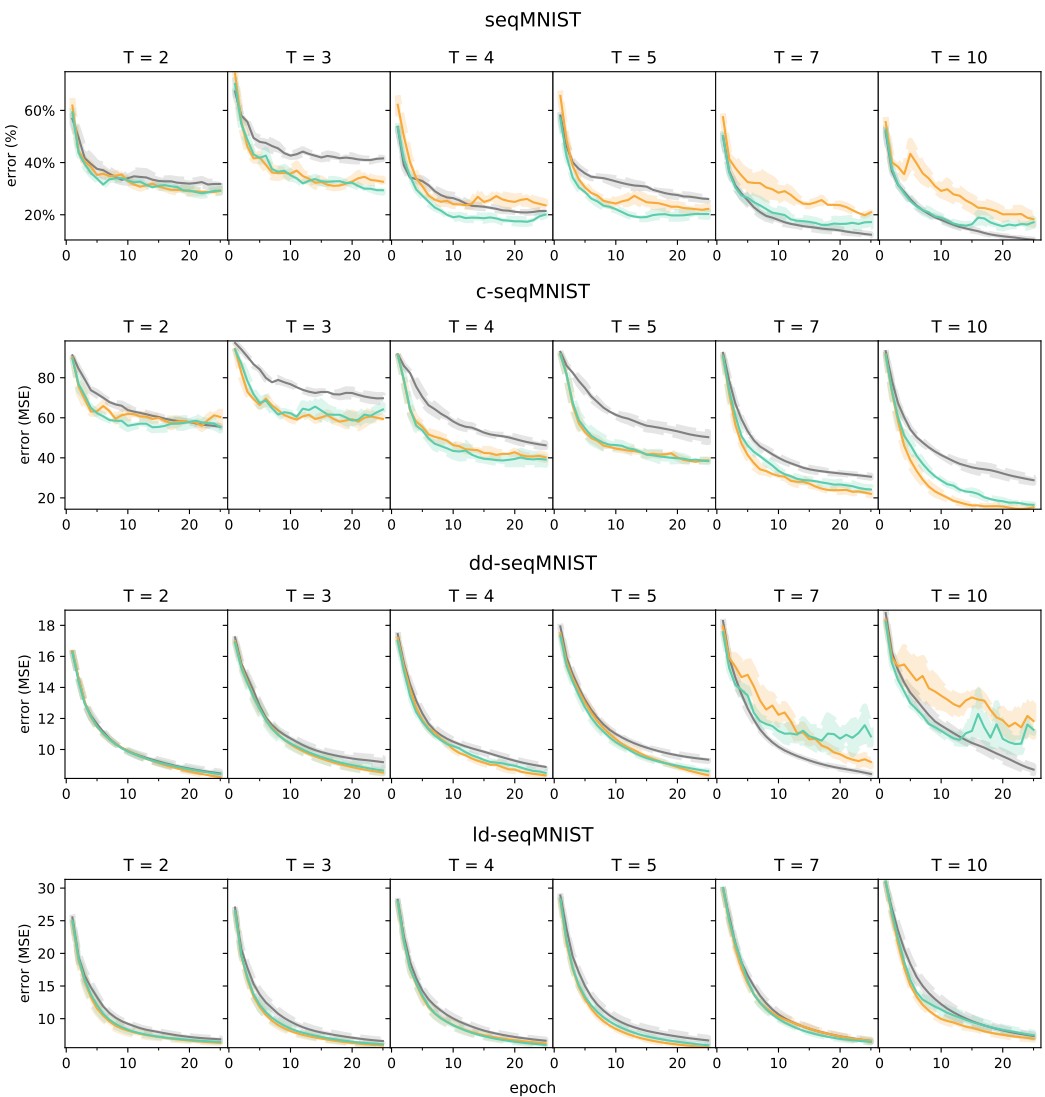

Figure S10: Learning curves for all seqMNIST based tasks across different truncation values. Model colours as in the main text (LSTM grey; CC-DNI orange; sCC-DNI cyan), with the default parameters of a $5/1$ divergence (300 GCs) in the DNI models and $K = 4$ for sCC-DNI.

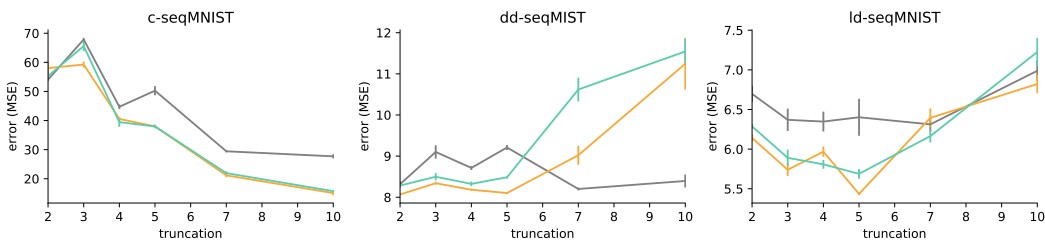

Figure S11: Related to Fig. S10. Average performance over last five epochs (21-25) for c-seqMNIST, dd-seqMNIST, ld-seqMNIST tasks across truncation sizes for LSTM (grey), cc-DNI (orange), sCC-DNI (cyan). See Fig. 3c for (standard) seqMNIST.

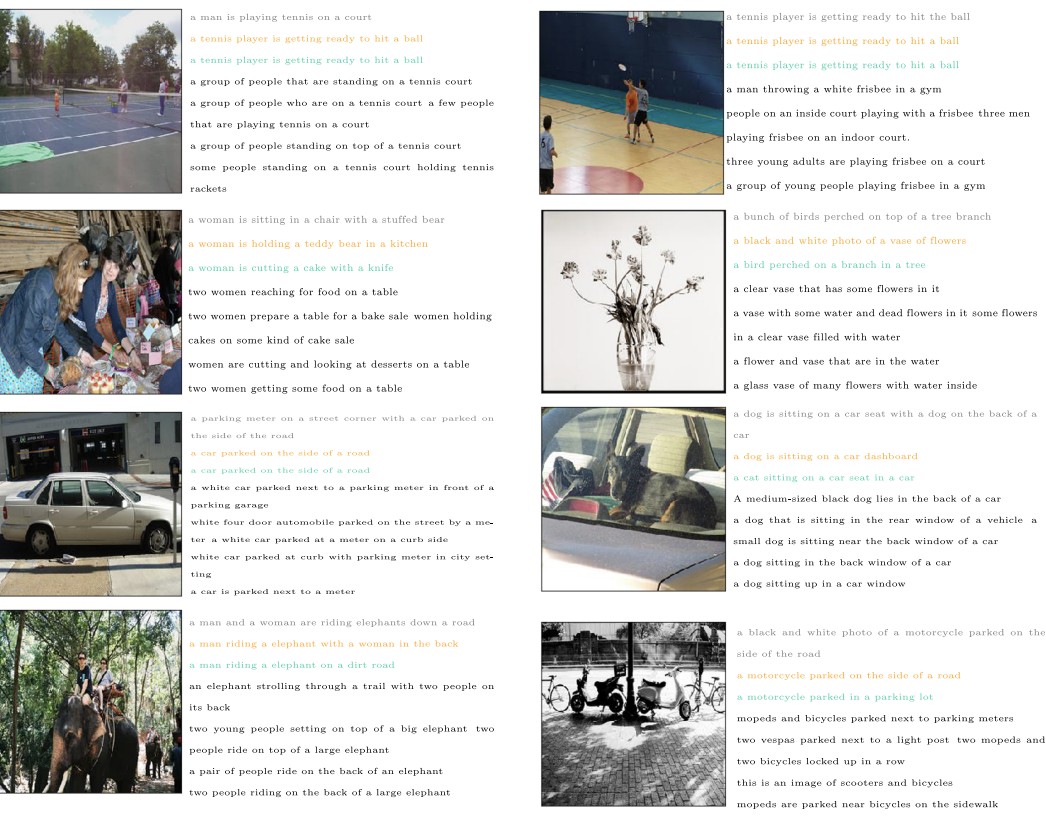

Figure S12: Example images from the validation set with corresponding model captions (LSTM (grey); CC-DNI (orange); sCC-DNI (cyan)) and gold standard captions (black). Here we show a combination of examples of how the models describe the presented image. In some case all or some models fail to give an accurate description of the image. In other cases all models are able to an accurate caption describing the image, with each model displaying subtle differences in the generated captions.

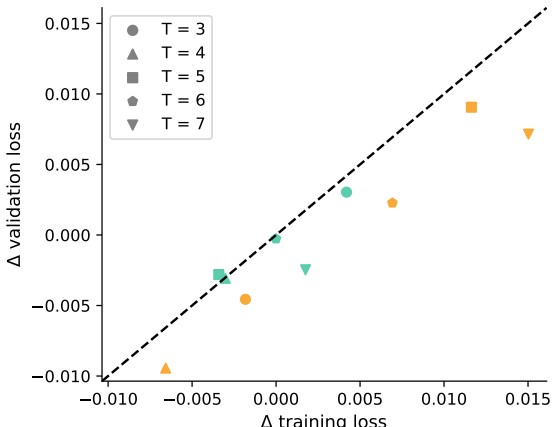

Figure S13: Generalisation of CC-DNI (orange) and sCC-DNI (cyan) for truncation values $T$ from 3 to 7. The change in loss is computed with reference to the LSTM (i.e. (s)CC-DNI - LSTM). Training loss is calculated *after* training (with dropout disabled) to enable fair comparison with final validation performance.

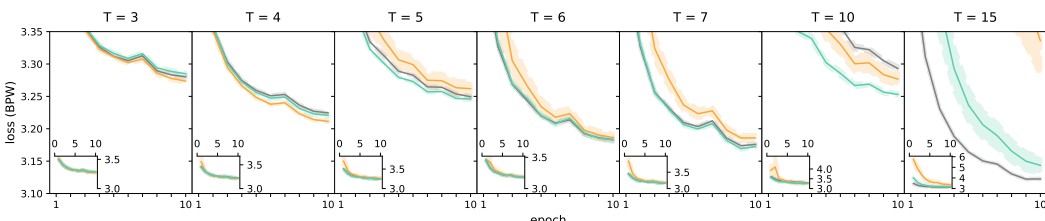

Figure S14: Validation learning curves in bits per word (BPW) for LSTM (grey), CC-DNI (orange) and sCC-DNI (cyan) across different truncation sizes $T$ (see Fig. 4). BPW range restricted to enable comparison between truncation values; full curves are shown in inset. The surprising performance for $T = 10$ is likely due to how the sequence is divided into truncations. With a strong majority of (gold standard) caption lengths between 11-15 (mean $\sim 13$, the sequence will often be divided into two uneven truncations, perhaps making BPTT difficult for the LSTM. However, in the case of two truncations a synthetic gradient will only be required once (in between the two truncations) and is analogous to the "easier" job of a synthesiser predicting gradients for a feedforward network, explaining the particular relative improvement seen for the CC-DNI models in this case.

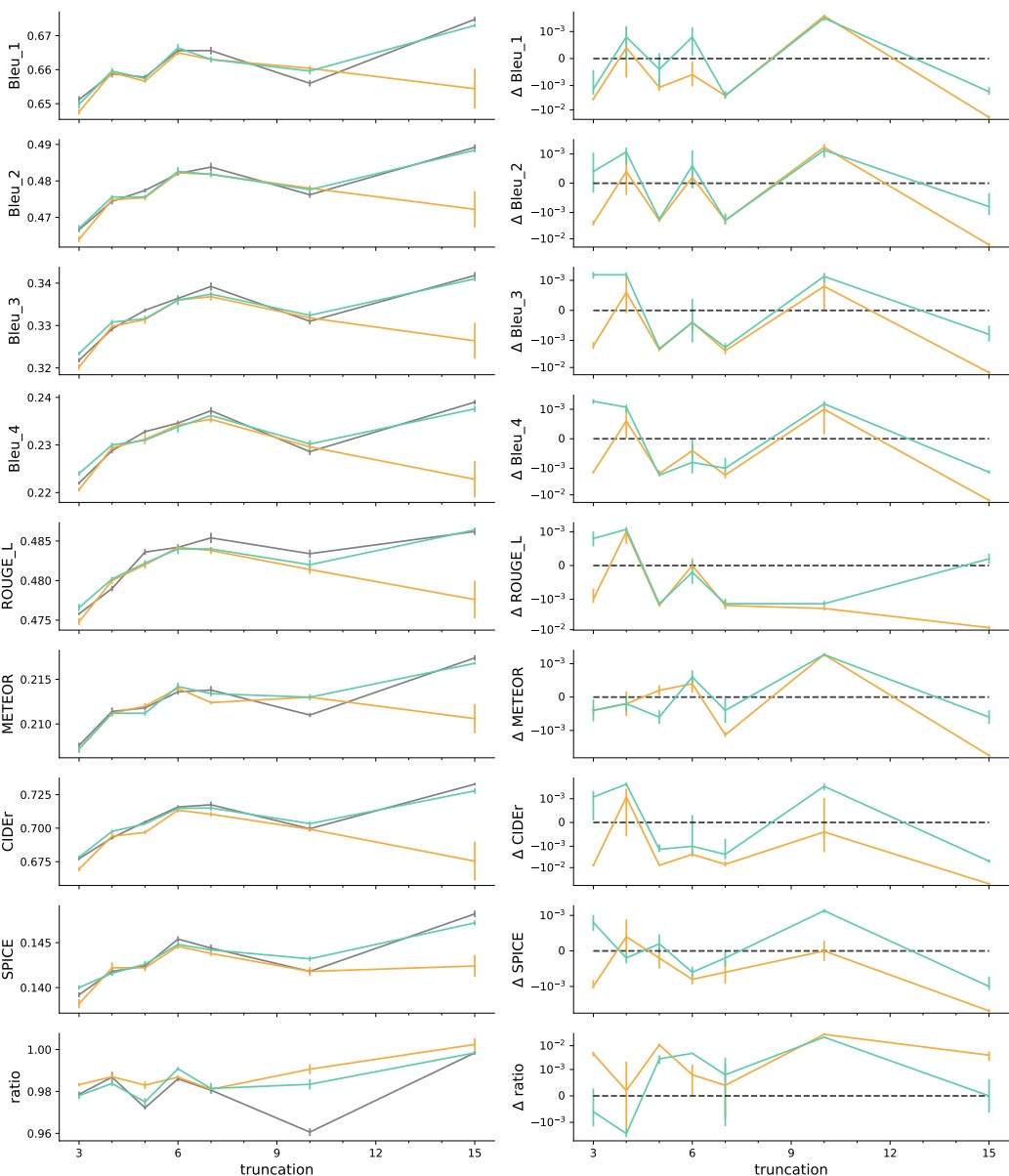

Figure S15: Evaluation of model-generated captions across truncation sizes for metrics (in order as shown) BLEU_1, BLEU_2, BLEU_3, BLEU_4, Rouge-L, METEOR, CIDEr, SPICE. The caption length *ratio* $C_m/C_{gs}$, where $C_m$ is the length of the model-generated caption and $C_{gs}$ is the length of the corresponding gold standard caption of closest length, is also shown.

