# OpenReview forum: "Cortico-cerebellar networks as decoupled neural interfaces"
_ICLR.cc/2021/Conference — Reject_

### Official Review · AnonReviewer2 · 2020-10-21
**Interesting idea but relatively weak connection to cerebellum**

**Rating:** 3
**Confidence:** 3

**Review:**

The authors proposed that cerebellum in the brain computes synthetic gradients, as used in decoupled neural interfaces (DNI), to enable learning in neural circuits without waiting for gradients to propagate backwards. The authors incorporated several architectural properties of biological cerebellum into their cerebellar model. They showed that a LSTM trained with such synthetic gradients can learn a variety of tasks, from motor reaching to caption generation. The paper is clearly written, and the link between DNI and cerebellum is a novel idea. However, the authors made little attempt to actually compare their model with experimental findings from cerebellum (except for the cerebellar properties built into the network), limiting its scientific impact. Meanwhile, it is not clear whether the cerebellum-inspired DNI provides concrete advantages over DNI proposed in Jaderberg 2017.

Major comments:
(1) The authors made little attempt to actually compare their model with experimental findings, or at least make concrete testable predictions.

Main results (Fig. 2-4) are mostly showing that their core models, CC-DNI and sCC-DNI, can successfully reduce losses across a variety of tasks, and do so better than heavily truncated BPTT gradients.

Except for the architectural features incorporated into the model, and some loose arguments that cerebellum is involved in sensory, motor, and cognitive tasks, the link between the model and biological cerebellum appears somewhat weak.

The authors claimed that “our work makes numerous predictions and opens the exciting door to explicit comparison…” without actually spelling out any key prediction. One prediction of the model is that when a task is very well-learned, the gradients (both real and predicted) should be close to 0, and cerebellum output neurons in the deep cerebellar nuclei should be somehow silent. Another prediction is that cerebellum lesioning should only impact learning, but not performance of well-learned tasks. I’m not sure these predictions are supported by empirical evidences.

I would feel much more comfortable supporting this manuscript if the authors provide more comparisons with experimental data, and make more concrete testable predictions.

(2) Critical questions about how real gradients are computed and transmitted to inferior olive is not answered.

For the brain, the backward lock may not be the most acute issue when searching for approximated gradient descent in brains. DNI relies on computing the real gradients, and using it to train the generation of synthetic gradients. Both computations are challenging in the brain. The authors completely circumvent this problem by saying it is “outside of the scope of the current paper”. However, I think this issue is critical for considering the feasibility of the proposed mechanism. For example, how can cerebellum learn to predict the gradient of individual cortical neurons when there are many fold less Purkinje cells than cortical neurons? There are of course more granule cells than there are cortical neurons, but in the model, granule cells are not the ones representing the synthetic gradient, \hat{g}_M, right?

Minor
(1) The references are at a number of places somewhere between inaccurate and incorrect. Here are a few that I noticed.

In the introduction, the authors wrote “These observations suggest that the cerebellum implements a universal function across the brain (Diedrichsen et al., 2019)”. However, if I’m not mistaken, the Diedrichsen review is arguing the exact opposite that cerebellum is not implementing a universal function.

In section 2.1.1, the authors wrote “Here we use LSTMs (Hochreiter and Schmidhuber, 1997) as a model of cortical networks, which have recently been mapped onto cortical microcircuit (Costa et al., 2017) ”. Costa 2017 provided a potential way to link cortical microcircuit to a LSTM-like structure. I don’t think it’s fair to say that LSTMs are "mapped" onto cortical microcircuits.

In section 2.1.2, the authors wrote “On the other hand Bellec et al. (2019) showed that temporal gradients as used in BPTT are equivalent to using eligibility traces that transmit gradient information forward in time”. The eligibility-trace-based algorithm proposed by Bellec 2019, namely e-prop, is not “equivalent” to back-prop. It is an approximation that works well empirically in the cases studied in that paper.

(2) Fig. 2 panels are mislabeled.

---

> ### Author Response · Authors · 2020-11-24
> **Response to AnonReviewer2**
>
> We thank the reviewer for the positive, constructive and detailed feedback. We believe we have now address the points raised. In particular, we have added three new figures, a new section (5) and extended the discussion to highlight predictions made by the model.
>
> 1. Predictions and comparison with experimental findings:
>
> (i) New Fig. 5: Shows that the experimentally observed cerebellar expansion and sparsity are good parameters for pattern recognition tasks, which predicts that the cerebellum might have evolved to help with these types of tasks. This point was previously in the SM but seems to have been missed, so we decided to bring it to the main text.
>
> (ii) New Fig. 6: Inspired by classical neuroscience ablation studies we performed ablation experiments to predict when are the cerebellar estimations most important. Our results show that as learning progresses the cerebellum becomes less important even impairing learning once the main network can easily learn the task on its own.
>
> (iii) New Fig. 7: We performed a correlation analysis to highlight changes in correlation structure between the cerebellum and the main network. We calculated correlations between the main network and the cerebellar module using a simple feedforward CC-DNI to make a cleaner illustration of the point (but similar results should hold for RNNs). Our results show a drop in correlations of granule cells more initially correlated with the main network, and an increase for granule cells that end up with a high correlation, consistent with recent experimental findings (see Fig. 6B in Wagner et al. 2019 Cell). In addition, we predict that such changes in correlations should be more evident when comparing the main network with the output cerebellar nuclei.
>
> (iv) Other predictions: We have added several other predictions to the discussion. "Moreover, the model also predicts which connections should project to GCs (source brain area activity) and the inferior olive (target brain area activity).  In addition, as demonstrated by Czarnecki et al. (2017) the CC-DNI model also predicts that without a cerebellum the neocortical representations should become less distributed across different brain areas. Furthermore, as shown by Jaderberg et al. (2017) these models enable a network to be trained in a fully decoupled fashion, which means that it can update asynchronously. Given that the brain is asynchronous this may be a fundamental benefit of having a cerebellar system.”
>
> 2. Gradients prediction: It is true that when approximating gradients as in our experiments, under the assumption that the task is learnt near perfectly (close to zero error), the model output will eventually become near silent. However, our framework is general and can be applied to predict any type of activity. In particular, in a biologically plausible model of backprop these is no longer going be the case as the model would simply predict (feedback) activity (e.g. Sacramento et al. NeurIPS 2018). To highlight this, we have used a simple feedforward DNI that predicts activity (Fig. 7 top) to show that when predicting activity the cerebellum will converge to some non-zero value which reflects the activity of the main network.
>
> 3. Purkinje/cortical numbers: It is true that there are more cortical neurons than Purkinje Cells. But it is also true that not all cerebellar predictions might be useful at a given point in time, so we postulate that the thalamus which is the gateway between the cerebellar output and the neocortex decides which signals should be sent through, potentially fixing some of the issues that we observe in the ablation study (I.e. That the cerebellum signals can sometimes impair learning). The reserve would happen via the Pons when deciding which signals to send to the cerebellum. This is a direction of research that we are very much interested in exploring in the future.
>
>
> Minor points:
>
> We apolagize for not being accurate when citing relevant papers. We agree and have fixed all the points raised. Regarding the review by Diedrichsen et al. they do contrast the universal and the modular view, but finish by saying “Looking across domains, we may ultimately discover a universal cerebellar transform”, so we thing this is a recent balanced review that is appropriate. However, we have also added more classical citations to support the idea of a universal cerebellar function.

---

### Official Review · AnonReviewer4 · 2020-10-26
**A clever mixture of existing ideas**

**Rating:** 5
**Confidence:** 5

**Review:**

The authors consider the architecture of the cerebellum as the predictive component of a decoupled neural interface (DNI). Using this framework, they perform experiments training networks with BPTT on several temporal tasks.

The paper is exceptionally clear and the experimental investigations are well-done.

However, it does not offer any new insight into either the cerebellum or DNI; rather it simply juxtaposes the details of two existing bodies of knowledge. The two statements that might most closely constitute new insight—that DNI is helpful on temporally challenging tasks and that sparsity-induced de-correlation can be helpful—were both established within their respective research domains. The logical induction that the authors make "predicting that the cerebellum is particularly important for more temporally challenging tasks" does not require the DNI to be established. That enforcing the architectural constraints of sparsity within a DNI might be helpful, although curious, does not constitute sufficiently extensive findings for a publication, and (although aided by) does not require connecting DNIs and the cerebellum.

Much like the platitude "the brain is like a computer" offers no new insight into either computers or brains, here, I do not believe that authors' investigations, although amusing to follow-along with, have added significant insight into either the cerebellum or DNIs. Therefore, I cannot offer strong support for acceptance.

---

> ### Author Response · Authors · 2020-11-24
> **Response to AnonReviewer4**
>
> We thank the reviewer for the positive feedback and points raised. We have revised the substantially  manuscript to address the points raised.
>
> 1. Lack of new insights: Our model architecture is mapped onto cerebro-cerebellar circuits to show similar deficits in motor and non-motor tasks to what has been observed in cerebellar-patients. To the best of our knowledge this is the first time that this has been demonstrated. This in turn means that it opens a new sub-field between cerebellar neuroscience and deep learning methods. However, we do agree that more insights should be included. We have added three new figures and one new section (5) with numerous predictions that the model makes regarding cerebellar function:
>
> (i) New Fig. 5: Shows that the experimentally observed cerebellar expansion and sparsity are good parameters for pattern recognition tasks, which predicts that the cerebellum might have evolved to help with these types of tasks. This point was previously in the SM but seems to have been missed, so we decided to bring it to the main text.
>
> (ii) New Fig. 6: We performed ablation experiments to predict when the cerebellar estimations are most important. Our results show that as learning progresses the cerebellum becomes less important even impairing learning once the main network with BPTT can solve the task on its own.
>
> (iii) New Fig. 7: We calculated correlations between the main network and the cerebellar module using a simple feedforward CC-DNI. Our key result is that only a subset of units become more correlated over learning, consistent with recent experimental findings (Wagner et al. 2019 Cell).
>
> (iv) We have also extended the discussion with more predictions: "Moreover, the model also predicts which connections should project to GCs (source brain area activity) and the inferior olive (target brain area activity).  In addition, as demonstrated by Czarnecki et al. (2017) the CC-DNI model also predicts that without a cerebellum the neocortical representations should become less distributed across different brain areas. Furthermore, as shown by Jaderberg et al. (2017) these models enable a network to be trained in a fully decoupled fashion, which means that it can update asynchronously. Given that the brain is asynchronous this may be a fundamental benefit of having a cerebellar system.”
>
> 2. Cerebellum-inspired insights for DL: There are many possible avenues to extend DNI models inspired by the cerebellum. We now highlight the functionally distinct modular structure of the cerebellum and the link to bootstrapped learning as used in DNIs. One of the key draw backs of DNIs is that they struggle to learn temporal gradients. Having multiple modules bootstrapping onto each other generalizes the ideas introduced by Jaderberg et al. 2017 and can in principle lead to DNI models that learn more quickly.

---

### Official Review · AnonReviewer1 · 2020-10-30
**Sginificant and original hypothesis linking ML and neuroscience; some clarity issues.**

**Rating:** 7
**Confidence:** 3

**Review:**

**General**

The paper presents a very intriguing hypothesis, and I believe that its publication will benefit the community and stimulate fruitful discussion. The model seems to offer a compelling and fairly novel explanation of cerebellar deficits (including non-motor) with a broad significance across the neuroscience and deep learning communities. That said, I think that there are opportunities for improving clarity and filling in details that will be lost on readers who aren't strongly familiarity with Jaderberg et al.

The paper currently presents the model primarily in the feedforward setting, and the results in the recurrent setting. This becomes confusing, since the two settings can be associated with different locking problems in neuroscience (i.e. bio-plausible alternatives to backprop vs. learning from delayed rewards / bio-plausible BPTT). For instance, the abstract and main text intro ("a given layer has to wait for all the layers above to finish computing its gradients") seems to propose a solution for feedforward bio-plausible backprop. In the results, however, it focuses primarily on the ability to learn from delayed signals with BPTT models. I think it may help to focus earlier on the recurrent setting with references to the delayed feedback problem.

In Section 2, the cortical network uses the backward synthesiser to avoid needing to wait for the loss signal - however, this seems to merely shift the problem since now the synthesiser will need to wait for the loss signal to train. Reading Jaderberg et al. (and the SM), it's clear that the synthesiser is instead continually trained on bootstrapped estimates. I think that a reference to bootstrapping in the main text (and a reference to the analogy with bootstrapped value functions in RL, as in Jaderberg) would make the model much clearer.

Since Jaderberg et al. have already shown that DNIs improve on truncated LSTMs, it seems like a more interesting comparison in the results would be DNI vs. CC-DNI.  If the authors are proposing that CC-DNI is a competitive deep learning approach than I would like to see something like the original DNI architecture included as a baseline. Otherwise, I would at least like to see a clearer description of the architectural differences between DNI (as previously implemented) and CC-DNI/sCC-DNI.

**Details**

- Do the predictions/consequences in Fig. 1, g, roughly correspond to dL/dh_i? If so it would help to make this explicit.
- Figure 2(f)(g) are labelled in the caption as (e)(f).
- Unlike Jaderberg et al., the paper combines update and backwards locking under the same label. A clarification, either in the SM or a footnote, would help.

---

> ### Author Response · Authors · 2020-11-24
> **Response to AnonReviewer1**
>
> We thank the reviewer for the enthusiastic and constructive feedback, which we have taken on board.
>
> 1. Focus on temporal feedback signals: The main point made by the reviewer is a point that we have tried hard to get right. We are introducing a general framework, but needed to focus on a specific aspect of this more general framework, which we decided to be the temporal domain as it is easier to demonstrate the benefits of DNI (note, however, that they are not unique to the temporal domain). In the revised version we have a few things to make our presentation clearer:
>
> (i) we have from the outset (abstract) made it clear that although our framework is general, we are focusing on temporal feedback (i.e. BPTT).
>
> (ii) we have change Figure 1 to reflect this. We now present the general framework, and then provide more details on exactly how we deal with temporal delayed feedback problem using BPTT (Fig. 1a,b).
>
>
> 2. Bootstrapping clarification: We totally agree that we should have been clearer about this. As pointed out by the reviewer a key element is the use of bootstrapping, which we make now explicit (Fig. 1c) and point out that this can be interpreted as a given cerebellar module and potentially other modules helping one another to speed up cerebellar learning.
>
>
> 3. DNI vs (s)CC-DNI: Our sparse CC-DNI model indeed provides an improvement on the current DNI as shown by our results. The only difference between the (non-sparse) CC-DNI model and the DNI is that the ratio of LSTM/DNI units closely follows what is observed in biology. We have now clarified this as follows: “This (our model) is different to DNI, in which the synthesizer contains a single hidden layer with the same number of units as LSTM”.
>
>
> 4. In addition, we have added three new figures and a new section (5) with specific model predictions:
>
> (i) New Fig. 5: Shows that the experimentally observed cerebellar expansion and sparsity are good parameters for pattern recognition tasks, which predicts that the cerebellum might have evolved to help with these types of tasks. This point was previously in the SM but seems to have been missed, so we decided to bring it to the main text.
>
> (ii) New Fig. 6: We performed ablation experiments to predict when are the cerebellar estimations most important. Our results show that as learning progresses the cerebellum becomes less important even impairing learning once the main network with BPTT can solve the task on its own.
>
> (iii) New Fig. 7: We calculated correlations between the main network and the cerebellar module using a simple feedforward CC-DNI. Our key result is that only a subset of units become more correlated over learning, consistent with recent experimental findings (Wagner et al. 2019 Cell).

---

### Author Response · Authors · 2020-11-24
**General response**

We would like to thank all the reviewers for the detailed, insightful and positive comments.

The main point raised was the lack of specific insights or predictions. We have addressed this directly by adding three experiments to the main manuscript, a new section and an extended discussion highlighting specific predictions, which we contrast with existing literature where possible. The fact that our predictions are consistent with existing observations, further supports our model.

Following the advice from R1 we have also revised the manuscript substantially to make it clear that although we introduce a general framework, in this paper we focus on the temporal feedback problem. We further clarified the use of bootstrapping in our model, which made us suggest that the numerous modules known to exist in the cerebellum may act as an efficient bootstrapping mechanism that generalizes the one introduced by Jaderberg et al. 2017.

Overall, we believe that the paper has been substantially improved after addressing the reviewer's comments.

---

### Decision · Program_Chairs · 2021-01-07
**Final Decision**

**Decision:**

Reject

**Comment:**

Reviewers split on this paper with one arguing that it is an intriguing and significant paper for both neuroscience and deep learning, whereas others argued that it fails to answer some key questions and stops short of offering testable predictions or novel findings. In particular Reviewer 2 questioned the limited experimental predictions and their experimental backing, as well as the plausibility of gradient calculations.  Reviewer 4 raised more fundamental concerns about the significance of the paper's contributions. All reviewers appreciated the paper's clarity. Overall, though, Reviewers 2 and 4 raised enough significant concerns that I cannot recommend acceptance.